# Root-specific theanine metabolism and regulation at the single-cell level in tea plants (*Camellia sinensis*)

Shijia Lin[1†], Yiwen Zhang[1†], Shupei Zhang[1], Yijie Wei[1], Mengxue Han[1], Yamei Deng[1], Jiayi Guo[1], Biying Zhu[1], Tianyuan Yang[1], Enhua Xia[1], Xiaochun Wan[1], William J Lucas[2], Zhaoliang Zhang[1]*

[1]State Key Laboratory of Tea Plant Biology and Utilization, Anhui Agricultural University, Hefei, China; [2]Department of Plant Biology, College of Biological Sciences, University of California, Davis, Davis, United States

**\*For correspondence:**
zhlzhang@ahau.edu.cn

[†]These authors contributed equally to this work

**Abstract** Root-synthesized secondary metabolites are critical quality-conferring compounds of foods, plant-derived medicines, and beverages. However, information at a single-cell level on root-specific secondary metabolism remains largely unexplored. L-Theanine, an important quality component of tea, is primarily synthesized in roots, from which it is then transported to new shoots of tea plant. In this study, we present a single-cell RNA sequencing (scRNA-seq)-derived map for the tea plant root, which enabled cell-type-specific analysis of glutamate and ethylamine (two precursors of theanine biosynthesis) metabolism, and theanine biosynthesis, storage, and transport. Our findings support a model in which the theanine biosynthesis pathway occurs via multicellular compartmentation and does not require high co-expression levels of transcription factors and their target genes within the same cell cluster. This study provides novel insights into theanine metabolism and regulation, at the single-cell level, and offers an example for studying root-specific secondary metabolism in other plant systems.

## eLife assessment

This study combines experimental and theoretical approaches to examine metabolites at the single-cell level in tea plants. The authors skilfully integrated various tools available for this type of research, and meticulously presented and illustrated every step of the survey. The overall quality of the work is **convincing**, and it represents an **important** contribution to our understanding of the compartmentalization of biosynthesis pathways.

## Introduction

Different root cell types produce diverse and complex secondary metabolites in response to environmental and developmental clues (*Hong et al., 2017*; *Shaw et al., 2021*); furthermore, such secondary metabolites can also participate in environmental adaptation and development (*Garcia-Lemos et al., 2020*; *Hartman et al., 2017*; *Santelia et al., 2008*; *Schulz-Bohm et al., 2018*; *Steppuhn et al., 2004*; *Wan et al., 2018*; *Xu et al., 2017*). As examples, the alkaloid nicotine is synthesized, specifically, in the root and is transported to the shoot for resistance to biotic stresses (*Steppuhn et al., 2004*; *Xu et al., 2017*). Root-synthesized flavonoids regulate root tip growth through affecting auxin transport and metabolism (*Santelia et al., 2008*; *Wan et al., 2018*). Legume roots secrete flavonoids as signaling agents to attract symbiotic bacteria, such as *Rhizobium* for nitrogen fixation (*Hartman et al., 2017*). In *Abies nordmanniana*, volatile organic compounds (e.g., propanal, g-nonalactone, and dimethyl

**Figure 1.** Process employed for single-cell RNA sequencing (scRNA-seq) of tea plant roots. (**A**) Tea seedlings were used for theanine detection and scRNA-seq. Tea plant roots (~2.0 cm in length from the root tip) were harvested and used for scRNA-seq. (**B**) Theanine contents of leaf, stem, cotyledon, and root. (**C**) Cross section of tea plant roots (from the root tip back ~2.0 cm). Co, cortex; Xy, xylem; Ph, phloem; Ca, cambium; Pe, pericycle; En, endodermis; EP, epidermis. Scale bar = 200 μm. (**D**) Flow chart of scRNA-seq analyses.

The online version of this article includes the following figure supplement(s) for figure 1:

**Figure supplement 1.** Summary of the tea plant root single-cell RNA sequencing (scRNA-seq) data.

disulfide) function to recruit certain bacteria or fungi, such as *Paenibacillus*. *Paenibacillus* sp. S37 produces high quantities of indole-3-acetic acid that can then promote plant root growth (*Garcia-Lemos et al., 2020*; *Schulz-Bohm et al., 2018*).

In many medicinal and beverage plants, root-synthesized secondary metabolites are critical quality-conferring compounds (*Bailly, 2021*; *Hu et al., 2019*; *Schmid et al., 2018*). Lobetyolin (LBT), a *Codonopsis pilosula* (known as Radix codonopsis or Dangshen) root-synthesized polyacetylene glycoside, exhibits activities against various cancers, notably, gastric cancer (*Bailly, 2021*). Polysaccharides and lectins isolated from *Pseudostellaria heterophylla* (known as Taizishen or Haiershen) roots have multiple pharmacological activities (*Hu et al., 2019*). Root-synthesized polyphenols and triterpenoid saponins are crucial quality compounds extracted from *Glycyrrhiza glabra* (known as Licorice) and are employed in the food and beverage industry (*Schmid et al., 2018*). Due to the importance of secondary metabolites, the biosynthesis, transport, and regulation of plant root-synthesized secondary metabolites have been extensively studied (*Fernie and Tohge, 2017*; *Wang et al., 2018a*). However, in most cases, whole root materials were employed with very few assessing secondary metabolite biosynthesis and regulation, at the single-cell level.

Tea plant (*Camellia sinensis* L.), a crop with high economic value, is a member of the eudicot family, *Theaceae*, has diverse and active secondary metabolic activities (*Yu et al., 2020*). Tea plants produce and accumulate extensive amounts of secondary metabolites, including theanine, catechins, and caffeine (*Zhao et al., 2020*). These secondary metabolites confer tea with both unique flavor and multiple health benefits (*Zhao et al., 2020*). Theanine, a non-protein amino acid, endows the 'umami' taste and relaxing effect of the tea infusion (*Lin et al., 2022*). It is the most abundant free amino acid in the tea plant, accounting for 1–2% dry weight of the tender shoots (*Lin et al., 2022*). It is primarily synthesized in tea plant roots and is then transported to the shoots, via the vasculature system (*Figure 1A, B*; *Lin et al., 2022*).

Tea plant roots acquire inorganic nitrogen, especially ammonium, from the soil and assimilate it into amino acids, such as glutamate (Glu), glutamine (Gln), and alanine (Ala) (*Lin et al., 2022*). Ala can be decarboxylated, by alanine decarboxylase (CsAlaDC), to produce ethylamine (EA) (*Bai et al., 2019*;

*Zhu et al., 2021*), which can then be combined with Glu, through the action of theanine synthase (CsTSI), to producetheanine (*She, 2022*; *Wei et al., 2018*). The high level of EA availability is likely a key factor in the elevated accumulation of theanine in tea plants (*Cheng et al., 2017*; *Zhu et al., 2021*). Importantly, in the tea plant, CsAlaDC and CsTSI are both indispensable for theanine synthesis (*Zhu et al., 2021*). Theanine synthesis is tightly regulated, at multiple levels, especially at the transcriptional level (*Lin et al., 2022*; *Guo et al., 2022*); CsMYB6 bound to the *CsTSI* promoter can regulate theanine synthesis (*Zhang et al., 2021c*). Furthermore, CsMYB40 and CsHHO3 bound to the *CsAlaDC* promoter can regulate theanine synthesis in 'accelerator' or 'brake' mode, respectively, in response to N levels (*Guo et al., 2022*). Theanine transporters, including the tonoplast-localized CsCAT2 and plasma membrane-localized CsAAPs, have been proposed to mediate in theanine storage within the root and root-to-shoot transport, respectively (*Dong et al., 2020*; *Feng et al., 2021*). In addition, in response to N status in tea plant roots, theanine may affect apoplastic $H_2O_2$ accumulation to regulate lateral root development (*Chen et al., 2023*). Taken together, these studies have partially revealed the mechanisms for theanine synthesis, transport, and regulation, but information on the specific root cell types involved in these processes is lacking.

Recently, with the in-depth study of biological structure and function, it has become increasingly clear that differences exist in gene expression levels between cells, even if they appear to be within the same cell population. Bulk RNA-seq can be employed to investigate the average level of gene expression within whole tissues, whereas single-cell RNA sequencing (scRNA-seq) can capture single cells, from whole tissues, to detect the heterogeneity of cells, and obtain information on gene expression within single cells (*Shaw et al., 2021*). Thus, scRNA-seq provides a powerful approach to elucidate the potential involvement of cell heterogeneity in secondary metabolism in tea plant roots. Currently, scRNA-seq has been performed in several plants, including *Arabidopsis thaliana* (*Ryu et al., 2019*), *Oryza sativa* (*Zhang et al., 2021a*), and *Zea mays* (*Nelms and Walbot, 2019*). Importantly, algorithms are available to decode the cellular regulatory codes from scRNA-seq datasets (*Becht et al., 2018*; *Haghverdi et al., 2015*; *Haghverdi et al., 2016*; *Trapnell et al., 2014*), and these databases can provide a critical foundation for analyzing the cellular-based information on gene expression (*Chen et al., 2021*; *Xu et al., 2022*).

In plants, scRNA-seq has yet to be used for studying root-specific secondary metabolism. In this study, using high-throughput scRNA-seq, we analyzed the cell heterogeneity of theanine metabolism and regulation in the tea plant root. Our findings offer important insights into the molecular mechanisms underlying theanine synthesis, transport, and regulation, and provide a basis for studies on root-specific secondary metabolism in other plant systems.

## Results
### Tea plant root cell clusters identified by scRNA-seq

Tea plant roots synthesize and accumulate large amount of theanine (*Figure 1A, B*). The tea plant root cell types, including cortical (Co), xylem (Xy), phloem (Ph), cambial (Ca), pericycle (Pe), endodermal (En), epidermal (Ep), and root hair (RH), are shown in *Figure 1C*. To understand the cellular characteristics of theanine biosynthesis in the tea plant root, protoplasts were isolated from seedling roots (from the root tip back ~2.0 cm) of the tea plant cultivar 'Shuchazao' (*Figure 1A*). These protoplasts were subjected to scRNA-seq, using the commercial 10× Genomics platform (*Figure 1D*), and the sequencing data were pre-filtered, at the cell and gene levels, resulting in a pool of 10,435 cells (*Figure 1—figure supplement 1*; *Supplementary file 1* and *Supplementary file 2*). The *t*-distributed stochastic neighborhood embedding (*t*-SNE) algorithm was employed to visualize local similarities (*Haghverdi et al., 2015*), resulting in these 10,435 cells being grouped into eight cell clusters (*Figure 2A*).

The cell types within these eight clusters were assessed by an analysis of homologs of cell-type marker genes according to protein sequence similarity, confirmed in other scRNA-seq studies, or in the model plant *A. thaliana*. In *A. thaliana*, GLABRA3 (GL3) is involved in epidermal cell fate specification (*Bernhardt et al., 2003*). In our scRNA-seq, *CsGL3* transcripts were detected in Cluster 2, indicating the presence of epidermal cells (*Figure 2—figure supplement 1A*). The *SCARECROW* (*SCR*) gene is required for the asymmetric division of the cortex/endodermis initial (CEI) cells, and its expression is confined to the endodermis, CEI cell, and the quiescent center (QC) (*Dong et al., 2021*). In our

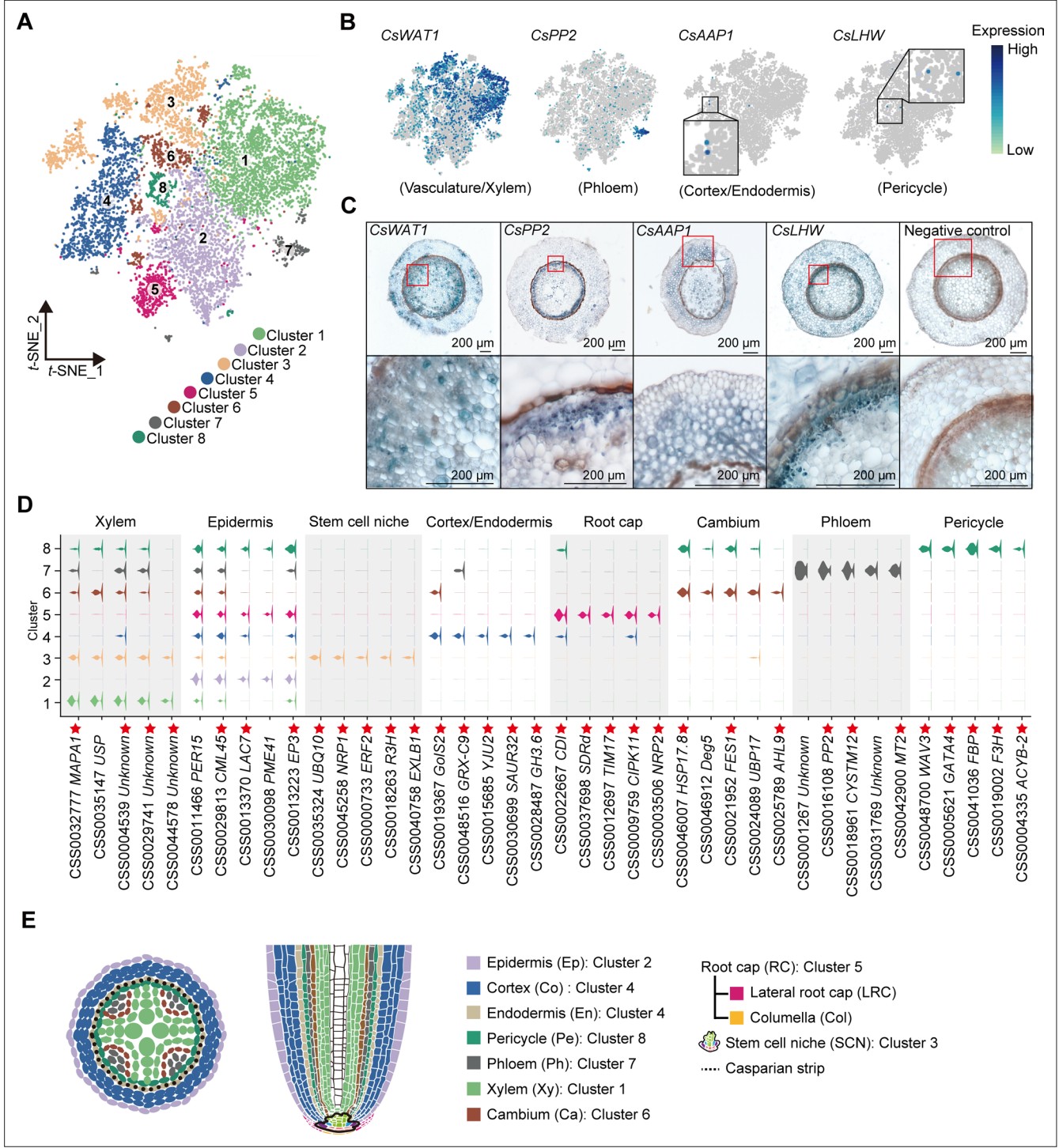

**Figure 2.** Annotation of cell clusters based on single-cell RNA sequencing (scRNA-seq) analysis of tea plant roots. (**A**) *t*-Distributed stochastic neighborhood embedding (*t*-SNE) visualization plot of 10,435 tea plant root cells showed that they were grouped into eight cell clusters. Each dot denoted a single cell. (**B**) *t*-SNE visualization of cell-type marker genes of root cells. Color bar indicates gene expression level. (**C**) *In situ* reverse transcription polymerase chain reaction (*In situ* RT-PCR) of *CsWAT1*, *CsPP2*, *CsAAP1*, and *CsLHW*. The blue areas of sliced tissue represent regions where genes are expressed. The red box represents the magnified areas of root sections. Scale bar = 200 µm. (**D**) Violin plot showed the expression patterns for the top 5 marker genes of each cell cluster. The height of violin represents the gene expression level, and the width represents the proportion of cells expressing in the cluster. Asterisks represent homologous *Arabidopsis thaliana* marker genes that can be found in PlantscRNAdb. (**E**) Schematic of the tea plant root anatomy, cell types, and their associated clusters.

The online version of this article includes the following figure supplement(s) for figure 2:

*Figure 2 continued on next page*

*Figure 2 continued*

**Figure supplement 1.** Cell cluster annotation.

**Figure supplement 2.** *t*-Distributed stochastic neighborhood embedding (*t*-SNE) visualization of Clusters 1 and 2 top 10 marker genes.

**Figure supplement 3.** *t*-Distributed stochastic neighborhood embedding (*t*-SNE) visualization of Clusters 3 and 4 top 10 marker genes.

**Figure supplement 4.** *t*-Distributed stochastic neighborhood embedding (*t*-SNE) visualization of Clusters 5 and 6 top 10 marker genes.

**Figure supplement 5.** *t*-Distributed stochastic neighborhood embedding (*t*-SNE) visualization of Clusters 7 and 8 top 10 marker genes.

tea plant root data, *CsSCR* transcripts were detected primarily in Clusters 3 and 4 (*Figure 2—figure supplement 1A*), suggesting that cortical and endodermal cell division occurs in these two clusters. However, as *CsAAP1* transcripts were detected in Cluster 4 (*Figure 2B*), and the gene encodes for AMINO ACID PERMEASE 1, which generally mediates in uptake of amino acids, into the cortex and endodermis, this cluster likely contained a population of cortical or endodermal cells.

The stele system in the root is comprised of the pericycle and vasculature, in which *LONESOME HIGHWAY* (*LHW*) has been established as a pericycle marker gene (*De Rybel et al., 2013*). As the *CsLHW* transcripts were detected in Cluster 8, this cell cluster likely contains a pericycle cell population (*Figure 2B*). The *WALLS ARE THIN LIKE 1* (*WAT1*) and *IRREGULAR XYLEM 9* (*IRX9*) are characterized vasculature and xylem development marker genes, respectively (*Li et al., 2021*). Hence, the high level of *CsWAT1* transcripts detected in Clusters 1 and 3 (*Figure 2B*), along with *CsIRX9* transcripts present in Cluster 3 (*Figure 2—figure supplement 1A*) support the notion that these two clusters represent vasculature or xylem-related cells in the tea plant root. Finally, the *PHLOEM PROTEIN 2* (*PP2*) gene participates in phloem development and is a well-characterized phloem marker (*Dinant et al., 2003*) and the presence in Cluster 7 of abundant *CsPP2* transcripts is consistent with phloem cells being located within this Cluster 7 (*Figure 2B*).

To confirm the annotation of these cell clusters, we performed *in situ* reverse transcription polymerase chain reaction (*in situ* RT-PCR) analyses. These assays indicated expression of the following genes in their respective cell types: *CsWAT1* in the vasculature (including pericycle, xylem parenchyma, and cambium); *CsPP2* in the protophloem; *CsAAP1* mainly in the cortex and endodermis; and *CsLHW* in the pericycle (*Figure 2C*). These findings were consistent with the above annotation of these cell clusters (*Figure 2B, C*).

Given the lack of more marker genes in tea plants, we next applied an unbiased approach, developed by *Denyer et al., 2019*, in which cluster-enriched genes are used as markers to define cell cluster identity. Cluster-enriched genes are defined as those being expressed, at significantly higher level, in each cluster, than in all other clusters, and expressed in ≥10% of cells within this specific cluster. The information on the cell-specific expression of these cluster-enriched genes was confirmed in *A. thaliana*. Therefore, we analyzed expression patterns of the top 10 cluster-enriched genes (*Figure 2D*, *Figure 2—figure supplements 1–5*, *Supplementary file 3*). Here, we could establish enrichment of *LACCASE 7* (*CsLAC7*) and *PECTIN METHYLESTERASE 41* (*CsPME41*) in Cluster 2, *ASPARAGINE RICH PROTEIN1* (*CsNRP1*) in Clusters 3 and 5, and *MITOCHONDRIAL IMPORT INNER MEMBRANE TRANSLOCASE subunit TIM17* (*CsTIM17*) in Cluster 5 (*Figure 2D*). Based on these scRNA-seq results from *A. thaliana*, and previous studies showing that *AtLAC7* and *AtPME41* are expressed mainly in epidermal cells, *AtNRP1* functions as a root meristem maker, including stem cell niche (SCN) and root cap (RC), and *AtTIM17* is expressed in the RC (*Passarinho et al., 2001*; *Zhu et al., 2006*; *Figure 2—figure supplement 1C*), we assigned Clusters 3 and 5 as SCN and RC cells, respectively.

Taken together, according to the above results, Clusters 1, 2, 4, and 7 were annotated as xylem, epidermal, cortex/endodermal, and phloem cells, respectively (*Figure 2E*). In addition, Cluster 8 was annotated as pericycle cells (*Figure 2E*), whereas Cluster 6 included primarily procambium or transition cells during differentiation (*Figure 2E*), and Cluster 3 was defined as the SCN, which serves as the cluster containing founder cells for differentiation (*Figure 2E*).

## Differentiation trajectories of tea plant root cells supported cell cluster annotations

During root cell differentiation, all root cell types originate from stem cells generated by the root apical meristem (RAM). In the RAM, stem cells, including stelar initial (SI) cells, CEI cells, epidermis/lateral root cap initial (ELI), cells and columella initial (CI) cells, surround the QC (*Efroni et al., 2016*;

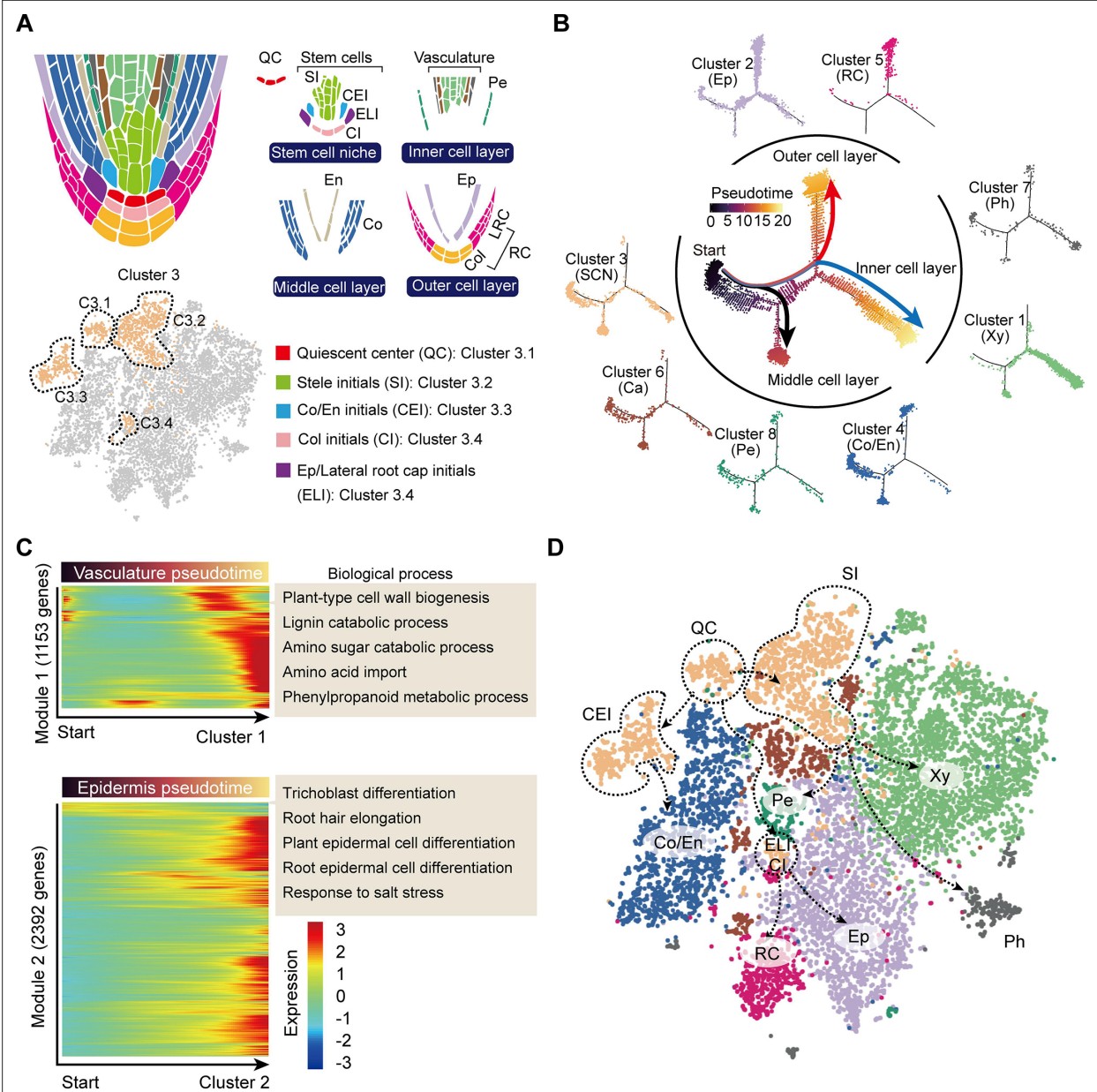

**Figure 3.** Trajectory of root cell differentiation in the tea plant root. (**A**) Schematic representation of root meristem organization (in median longitudinal section). *t*-Distributed stochastic neighborhood embedding (*t*-SNE) visualization and cell-type annotation of Cluster 3 (bottom). (**B**) Simulation of the successive differentiation trajectory of the stem cell niche (SCN) over pseudo-time. Each dot denotes a unique cell. Pseudo-time analysis of all 10,435 tea plant root cells (center), and the detailed distribution of eight cell clusters along pseudo-time trajectory (outer). Colors of the dots represent the pseudo-time score. Red and blue lines mark major differentiation trajectories, black lines mark minor differentiation trajectories. (**C**) Heatmap illustrating the expression patterns of differential genes, along the pseudo-time trajectory, during cell differentiation of the vasculature and epidermis. Each row represents one gene. Color bars indicate the relative expression levels. Biological processes are given on the right. (**D**) Proposed tea plant roots cell differentiation trajectories for cell clusters shown on the *t*-SNE visualization plots. Dotted arrows represent assumed cell differentiation routes.

The online version of this article includes the following figure supplement(s) for figure 3:

**Figure supplement 1.** Uniform manifold approximation and projection (UMAP) visualization of eight clusters and differentiation trajectory atop the UMAP.

*Figure 3A*). The QC cells are required for specification of the SCN and maintaining the undifferentiated state of the stem cell initials. Stem cells continuously undergo asymmetric division to produce daughter cells, which then begin to differentiate into outer, inner, and middle cell layers (*Efroni et al., 2016*; *Figure 3A*). We noticed that Cluster 3 could probably be divided into 4 sub-cell states,

C3.1, C3.2, C3.3, and C3.4 (*Figure 3A*), with C3.1 expressing the QC marker gene, *WUS-RELATED HOMEOBOX 5* (*CsWOX5*, *Kong et al., 2015*), C3.2 expressing the xylem development gene *CsIRX9*, C3.3 expressing the cortex and endodermal cell fate decision gene, *CsSCR*, and C3.4 being close to the Cluster 2 epidermal cell population (*Figure 3A*, *Figure 2—figure supplement 1A*). Thus, these four Cluster 3 sub-cell states were defined as QC, SI, CEI, and ELI cells, respectively (*Figure 3A*).

The differentiation trajectories of these clusters were next assessed to confirm our annotation of the grouped cell clusters. To this end, we conducted pseudo-time analysis by ordering cells of these clusters along a reconstructed trajectory (*Figure 3B*, *Figure 3—figure supplement 1*). This analysis indicated that Cluster 3 was primarily located at the start of the differentiation trajectories, and that the trajectories of these cells bifurcated into two major and one minor branches (*Figure 3B*). Here, the two major branches reflected differentiation trajectories of the ELI and SI cells from the SCN (Cluster 3), with the ELI cells differentiating into the outer cell layer (red branch) and Clusters 2 (epidermal cells) and 5 (RC cells) located at the end of this divergence (*Figure 3B*). The SI cells differentiated to give an inner cell layer (blue branch): Clusters 7 (phloem cells) and 1 (xylem cells) were arranged successively (*Figure 3B*). The minor branch resulted in CEI cell trajectories that differentiated to give a middle cell layer (black branch); Cluster 4 (cortex and endodermal cells) was located at the end of this divergence. Importantly, Clusters 4 (cortex and endodermis), 6 (cambium), and 8 (pericycle) cells were also located within the region where cell differentiation was initiated, thus suggesting these three cell clusters can differentiate into other cell types (*Figure 3B*). Indeed, the cambium differentiates into the xylem and the phloem, especially in woody plants *Li et al., 2021*; endodermal cells assume a stem cell identity to generate a new epidermal layer *Efroni et al., 2016*; the founder cells, derived from the pericycle, can form lateral roots (*Péret et al., 2009*). The results of our pseudo-time analyses matched well with these earlier established findings.

We next constructed a differentiation heatmap of xylem and epidermal cells (*Figure 3C*). Consistently, highly expressed genes in Cluster 1 (xylem cells) were most enriched in cell wall biogenesis, lignin catabolic process, amino sugar catabolic process, etc. (*Figure 3C*, *Supplementary file 4*). Meanwhile, genes highly expressed in Cluster 2 (epidermal cells) were enriched in trichoblast differentiation, root hair elongation, plant epidermal cell differentiation, etc. (*Figure 3C*, *Supplementary file 4*). These in-depth differentiation trajectories of root cells further indicated reliability of the cell cluster annotations.

Finally, a differentiation map of tea plant root cells was constructed, based on the pseudo-time analyses (*Figure 3D*). This map was generated to assist in developing an understanding of tea plant root development, to provide a basis for studying secondary metabolism, in tea plant roots, at the single-cell level.

## Cell heterogeneity of N transport and metabolism in tea plant roots

In tea plant roots, theanine accounts for approx. 60–80% of the total free amino acids (*Yang et al., 2020*). Nitrogen (N) uptake and transport, in the form of nitrate and ammonium, and N assimilation into amino acids, are prerequisites for theanine biosynthesis (*Yang et al., 2020*; *Figure 4A*). We therefore investigated the cell heterogeneity of these processes by analyzing the scRNA-seq data.

The NITRATE TRANSPORTERS (CsNRTs) and AMMONIUM TRANSPORTERS (CsAMTs) function in N uptake in tea plants (*Zhang et al., 2021b*; *Zhang et al., 2020*; *Figure 4A*). The heatmap of relative expression revealed that most *CsNRTs* and *CsAMTs*, including *CsAMT1.1*, *CsAMT1.2*, *CsNRT1.1*, *CsNRT1.5*, *CsNRT1.7*, *CsNRT3.1*, and *CsNRT3.2*, were expressed primarily in Cluster 4 (cortex and endodermal cells); *CsAMT1.2* and *CsNRT1.5* were also highly expressed in Cluster 7 (phloem cells); *CsAMT3.1* was expressed primarily in Clusters 8 (pericycle cells) and 6 (cambium cells); *CsAMT3.1* and *CsNRT3.2* were relatively specifically expressed in Cluster 2 (epidermal cells); *CsNRT2.4* was highly and specifically expressed in Cluster 3 (SCN) (*Figure 4B*). At the same time, our *in situ* RT-PCR results for *CsAMT1.1*, *CsAMT3.1*, *CsNRT1.1*, and *CsNRT3.2* confirmed these scRNA-seq findings (*Figure 4B, C*). Furthermore, these results were similar to previous reports in *A. thaliana* and rice (*Wang et al., 2018a*; *Wang et al., 2018b*).

Intracellular $NO_3^-$ is assimilated in the cytoplasm into $NO_2^-$, by NITRATE REDUCTASE (NR), which then enters the plastids where it is further assimilated into $NH_4^+$, by NITRITE REDUCTASE (NiR). Subsequently, $NH_4^+$ enters the amino acid metabolism pathway (*Galili et al., 2016*; *Xu et al., 2012*; *Figure 4A*). Our scRNA-seq assays revealed that expression of *CsNR1*, *CsNR2*, *CsNiR1*, and *CsNiR2*

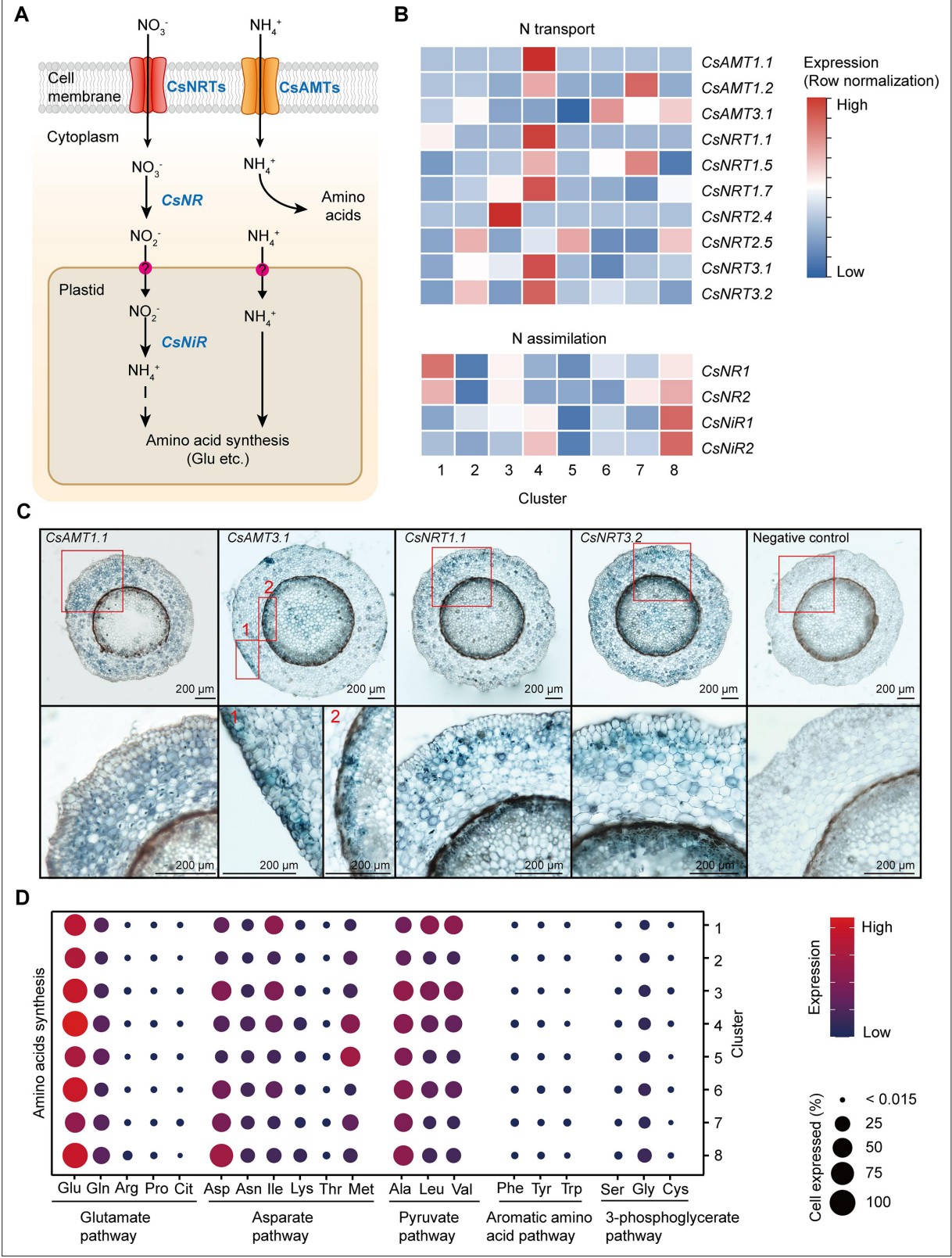

**Figure 4.** Cell heterogeneity of nitrogen (N) transport and assimilation in tea plant roots. (**A**) Schematic of N transport and metabolism pathway in tea plant roots. Membrane-located transporter NRTs and AMTs uptake nitrate and ammonium into root cells, respectively, and subsequently $NO_3^-$ is assimilated into $NH_4^+$ and then metabolized into various amino acids. (**B**) Heatmap shows cell cluster expression patterns of N transport and assimilation genes in tea plant roots. Row normalization for gene expression according to 'normalized' mode. (**C**) *In situ* RT-PCR of *CsAMT1.1*, *CsAMT3.1*, *CsNRT1.1*,

*Figure 4 continued on next page*

*Figure 4 continued*

and *CsNRT3.2*. The blue areas of sectioned tissues represent regions where genes are expressed. The red boxes represent the magnified areas of root sections shown on the right. Scale bar = 200 μm. (**D**) Expression of amino acid synthesis pathway genes in different cell clusters. Color bars indicate the expression levels of amino acid synthesis pathway genes. Dot size indicates the percentage of amino acid synthesis genes expressed in a cell cluster.

was more active in Cluster 8 (pericycle cells) (*Figure 4B*); *CsNR1* and *CsNR2* expression was also relatively high in Cluster 1 (xylem cells). These data supported the notion that N assimilation occurs primarily within stelar cells in tea plant roots.

Previously, we grouped amino acid metabolism into five pathways (P1–P5), based on their metabolic relationships (*Yang et al., 2020*): glutamate pathway (P1: Glu, Gln, arginine [Arg], proline [Pro], citrulline [Cit]); asparate (Asp) pathway (P2: Asp, isoleucine [Ile], lysine [Lys], threonine [Thr], and methionine [Met]); pyruvate pathway (P3: Ala, leucine [Leu], and valine [Val]); aromatic amino acid pathway (P4: phenylalanine [Phe], tyrosine [Tyr], and tryptophan [Trp]); and the 3-phosphoglycerate pathway (P5: serine [Ser], glycine [Gly], and cysteine [Cys]) (*Figure 4D*, *Supplementary file 5*). Theanine is synthesized from Glu- and Ala-derived EA (*Lin et al., 2022*).

Based on the scRNA-seq analysis, we observed that the expression level of genes encoding for Glu biosynthetic pathway enzymes was the highest (especially in Clusters 1, 3, 4, 6, and 8) within these five pathways (*Figure 4D*, *Supplementary file 5*), suggesting that biosynthetic activity for Glu synthesis is highly active in the roots. Meanwhile, the expression level of genes encoding alanine biosynthetic pathway enzymes was also high in all these clusters, especially in Clusters 3 (SCN) and 8 (pericycle cells) (*Figure 4D*, *Supplementary file 5*). These findings indicated that Glu and Ala metabolism represent the major amino acid metabolism in tea plant roots. This is the basis for N metabolism into theanine in these roots.

## Cell types participating in theanine biosynthesis, storage, and transport in tea plant roots

The genes encoding enzymes in the theanine biosynthetic pathway, theanine transporters, and transcription factors (TFs) regulating theanine biosynthesis, were recently identified (*Dong et al., 2020*; *Feng et al., 2021*; *Guo et al., 2022*; *She, 2022*; *Wei et al., 2018*; *Zhang et al., 2021c*; *Figure 5A*). We analyzed the cell-specific expression patterns of these genes using our scRNA-seq data (*Figure 5B*, *Supplementary file 6*). Impressively, although *CsTSI* was highly expressed in all clusters (*Figure 5C*), it was expressed, more specifically, in Cluster 8 (pericycle cells) (*Figure 5B*). Here, our *in situ* RT-PCR assays also revealed that *CsTSI* is expressed primarily in the pericycle (*Figure 5D*). Consistently, Glu biosynthetic pathway genes, including *CsGOGAT1*, *CsGDH2*, and *CsGDH3*, also had a relatively high expression level in Cluster 8 (*Figure 5B*), although some *CsGOGAT* and *CsGDH* members were also relatively highly expressed in Clusters 1 (xylem cells), 2 (epidermis cells), 3 (SCN), 4 (cortex and endodermal cells), 5 (RC cells), 6 (cambium cells), and 7 (phloem cells) (*Figure 5B*).

The *CsTSI* and *CsAlaDC* are the most important genes for theanine synthesis (*She, 2022*; *Zhu et al., 2021*). However, we observed that *CsTSI* and *CsAlaDC* were not enriched in the same cluster. Different from *CsTSI*, *CsAlaDC* was most highly expressed in Cluster 1 (xylem cell) and was also relatively highly expressed in Cluster 6 (cambium cell) (*Figure 5B, C*). The *in situ* RT-PCR assays verified the expression pattern of *CsAlaDC* in the vasculature and cambium, and also showed that *CsAlaDC* was expressed, at a low level, in the pericycle (*Figure 5D*). Therefore, generally, *CsTSI* and *CsAlaDC* are co-expressed in the pericycle, where *CsTSI* is most highly expressed, whereas *CsAlaDC* is mostly highly expressed in the vasculature, where *CsTSI* has a lower expression level. These results suggest that the theanine biosynthesis pathway has a multicellular compartmentation characteristic, as EA would be synthesized primarily in the vasculature, whereas theanine would be more actively synthesized in the pericycle.

Generally, root-synthesized theanine is stored in the root cells, or transported from root to shoot (*Dong et al., 2020*; *Lin et al., 2022*). In this regard, CsAAPs (CsAAP1, CsAAP2, CsAAP4, CsAAP5, CsAAP6, and CsAAP8) and CsCAT2 were recently identified as theanine transporters (*Dong et al., 2020*; *Feng et al., 2021*). Among them, CsCAT2 probably mediates theanine storage in the vacuole, with CsAAP1 functioning in theanine loading into the xylem transpiration stream for root-to-shoot transport (*Dong et al., 2020*; *Feng et al., 2021*). Interestingly, *CsCAT2* was shown to be expressed most highly in the same cluster as *CsTSI*; that is, in Cluster 8 (pericycle cells) (*Figure 5B*). *CsCAT2* was

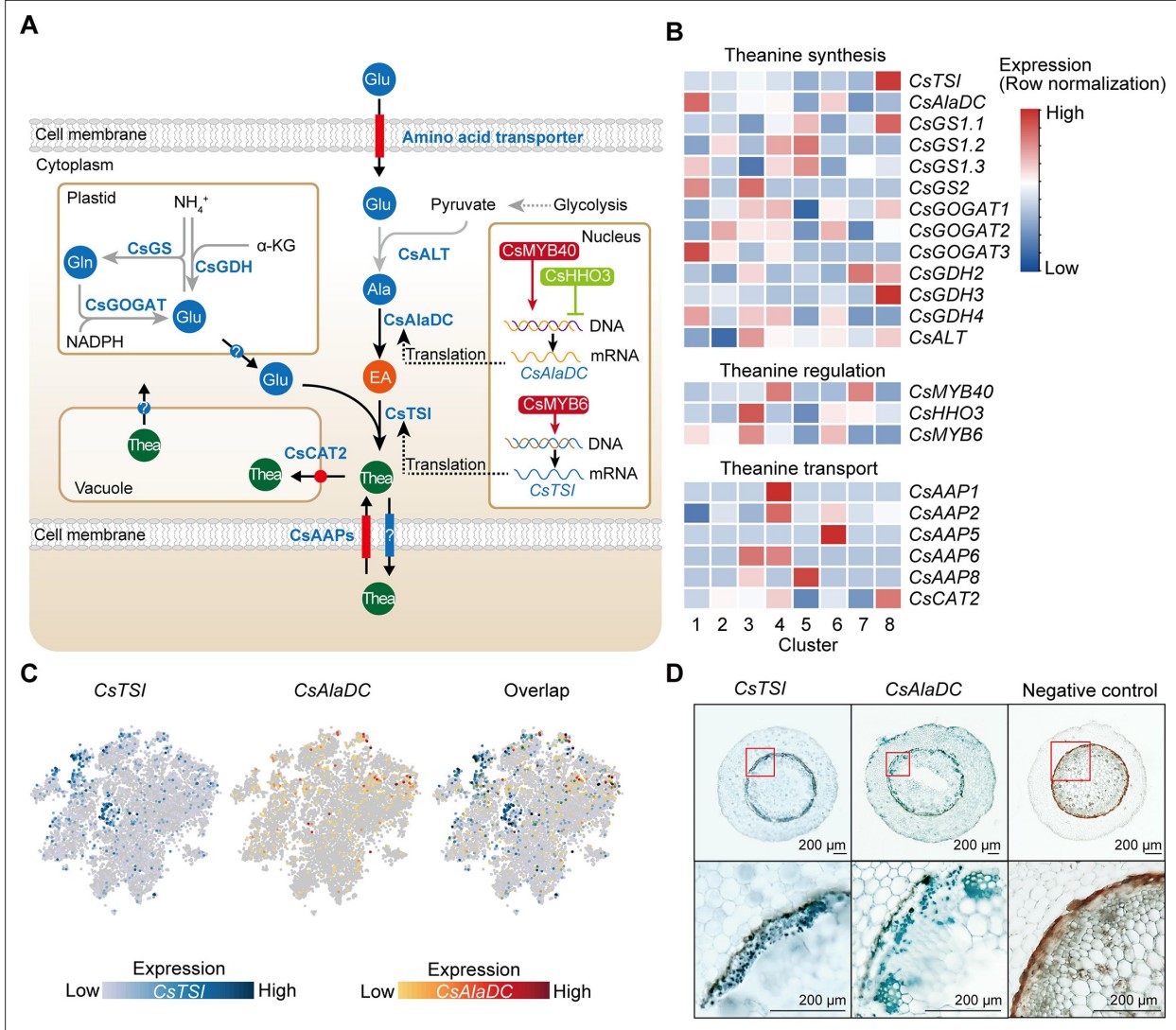

**Figure 5.** Cell heterogeneity of theanine synthesis, transport, and regulation in tea plant roots. (**A**) Model of the putative theanine synthesis, and regulation in tea plant root cells. Black full lines represent unique metabolic processes in tea plants; gray full lines represent common metabolic pathways in plants; dotted lines indicate the multi-step process. (**B**) Heatmap for cell cluster expression patterns of genes encoding theanine transporters CsAAPs, key enzymes, and transcription factors in the theanine metabolic pathway. Gene expression is presented in 'normalized' mode. (**C**) *t*-Distributed stochastic neighborhood embedding (*t*-SNE) visualization of *CsTSI*, *CsAlaDC*, and their overlap map. Blue color bar indicates *CsTSI* expression level, red color bar indicates *CsAlaDC* expression level in single-cell RNA sequencing (scRNA-seq). (**D**) Tissue localization of *CsTSI* and *CsAlaDC* using *in situ* RT-PCR. Blue signal indicates gene expression in cells. The red boxes represent magnified areas of root sections shown below. Scale bar = 200 μm.

The online version of this article includes the following figure supplement(s) for figure 5:

**Figure supplement 1.** Cell cluster assay of regulators and target genes based on published *Arabidopsis thaliana* data.

also relatively highly expressed in Cluster 4 (cortex and endodermal cells), suggesting that CsCAT2 may function in theanine storage in the pericycle, cortex, and endodermis. In contrast, *CsAAP1* was expressed primarily in Cluster 4 (cortex and endodermal cells). This is consistent with its proposed role in retrieving extracellular theanine for its transport through the endodermis prior to xylem loading (*Dong et al., 2020*; *Lin et al., 2022*). The expression of other *CsAAPs* was also shown to be highly cell cluster specific; for example, *CsAAP2* in Cluster 4 (cortex and endodermal cells), *CsAAP4* in Cluster 6 (cambium cells), *CsAAP6* in Clusters 3 (SCN) and 4 (cortex and endodermal cells), and *CsAAP8* in Cluster 5 (RC cells) (*Figure 5B*), suggesting cell-type-specific functions for these CsAAPs in theanine transport.

## Transcriptional regulation of theanine biosynthesis

Theanine biosynthesis is regulated at the transcriptional level, based on recent findings that TFs can control *CsTSI* or *CsAlaDC* expression to regulate its pathway (*Zhang et al., 2021a*; *Guo et al., 2022*). Here, Zhang et al. reported that CsMYB6, a root-specific expressed TF, promoted theanine biosynthesis through activation of *CsTSI* expression. Thus, we were surprised that *CsMYB6* was expressed in Clusters 3 (SCN), 6 (cambium cells), and 1 (xylem cells), rather than in Cluster 8 (the high *CsTSI* expression cell cluster), as shown in *Figure 5B*. A similar situation was also observed in terms of *CsAlaDC* transcriptional regulation. We previously identified CsMYB40 and CsHHO3 to be an activator and repressor of *CsAlaDC* expression, respectively, in response to N levels (*Guo et al., 2022*). Analysis of our scRNA-seq data indicated that *CsMYB40* and *CsHHO3* were not highly expressed in Cluster 1 (the high *CsAlaDC* expression cell cluster); *CsMYB40* was relatively highly expressed in Clusters 4 (cortex and endodermal cells) and 7 (phloem cells); and *CsHHO3* was relatively highly expressed in Clusters 3 (SCN) and 6 (cambium cells) (*Figure 5B*). Above regulatory relationships all have a common feature, the TFs and their target genes were highly expressed in different cell types.

Next, to further explore whether this regulatory relationship also exists in other plants, we analyzed expression of some TFs and their target genes in the model plant *Arabidopsis*, using published single-cell RNA-seq data (*Ryu et al., 2019*; *Wendrich et al., 2020*; *Zhang et al., 2019*; *Denyer et al., 2019*; *Jean-Baptiste et al., 2019*; *Shulse et al., 2019*; *Shahan et al., 2022*) and databases (Root Cell Atlas, https://rootcellatlas.org/; BAR, https://bar.utoronto.ca/#GeneExpressionAndProteinTools). By integrating findings from various studies (*Fukaki et al., 2006*; *Okushima et al., 2007*; *Stracke et al., 2007*; *Bustos et al., 2010*; *Huang et al., 2018*; *Liu et al., 2022a*; *Tang et al., 2022*), we observed that there is a hierarchical regulatory network, in which IAA14 functions as the upstream regulator (*Figure 5—figure supplement 1A*). Similar to the situation in tea plants, the regulators were not present in exactly the same cell types in which their target genes were highly expressed. For example, *AtARF7* and *AtARF19* were highly expressed in the cortex and stele, respectively, whereas their target genes *AtLBD16* and *AtLBD29* were highly expressed in endodermal cells (*Okushima et al., 2007*; *Figure 5—figure supplement 1B, C*); *AtPHR1* was highly expressed in root epidermal and pericyte cells, but its target gene *AtF3′H* was highly expressed in the cortex and *AtRALF23* was highly expressed in xylem cells (*Liu et al., 2022b*; *Tang et al., 2022*; *Figure 5—figure supplement 1B, C*). In this hierarchical regulatory network, some TFs have multiple target genes, and TFs participate in multiple biological processes (root development, metabolism regulation, and plant immunity) through regulation of different target genes (*Figure 5—figure supplement 1A*).

Based on these findings, we hypothesized that, for the regulation of theanine biosynthesis, TFs and target genes are not necessarily always highly expressed in the same cells. Moreover, the regulation of theanine biosynthesis is probably coupled with the regulation of other biological processes. In the theanine biosynthesis pathway, CsAlaDC evolved a tea plant-specific activity in catalyzing EA biosynthesis (*Cheng et al., 2017*), which is essential for theanine biosynthesis (*Zhu et al., 2021*). *CsAlaDC* is highly and specifically express in tea plant roots (*Zhu et al., 2021*), and its expression levels were shown to be highly correlated with theanine content in the roots of various tea plant cultivars, and further, was under TF regulation (*Zhu et al., 2021*). Thus, taking the transcriptional regulation of *CsAlaDC* as an example, we next analyzed the TFs that were co-expressed with *CsAlaDC* to test this notion.

## Identification of CsLBD37, a novel TF that co-regulates both theanine synthesis and lateral root development

In tea plant roots, both theanine biosynthesis and root development are regulated by N level and its form (*Yang et al., 2020*). Moreover, theanine is probably involved in N-regulated lateral root development (*Chen et al., 2023*). This implicates the presence of TFs that can co-regulate theanine biosynthesis and development in the tea plant root. To test this notion, we identified genes co-expressed with *CsAlaDC*, using our previous transcriptome data obtained for roots that were supplied with various N treatments, including without N (0 N), 1.4 mM nitrate ($NO_3^-$), 1.4 mM EA (EA), 1.4 mM ammonium ($NH_4^+$), or 1.4 mM nitrate plus ammonium (1:1) (CK) (*Supplementary files 7 and 8*). By filtering of these genes, using |correlation coefficients| ≥ 0.85 and p ≤ 0.05, we constructed a *CsAlaDC* co-expression network that consisted of 133 genes (*Figure 6A*, *Supplementary file 8*).

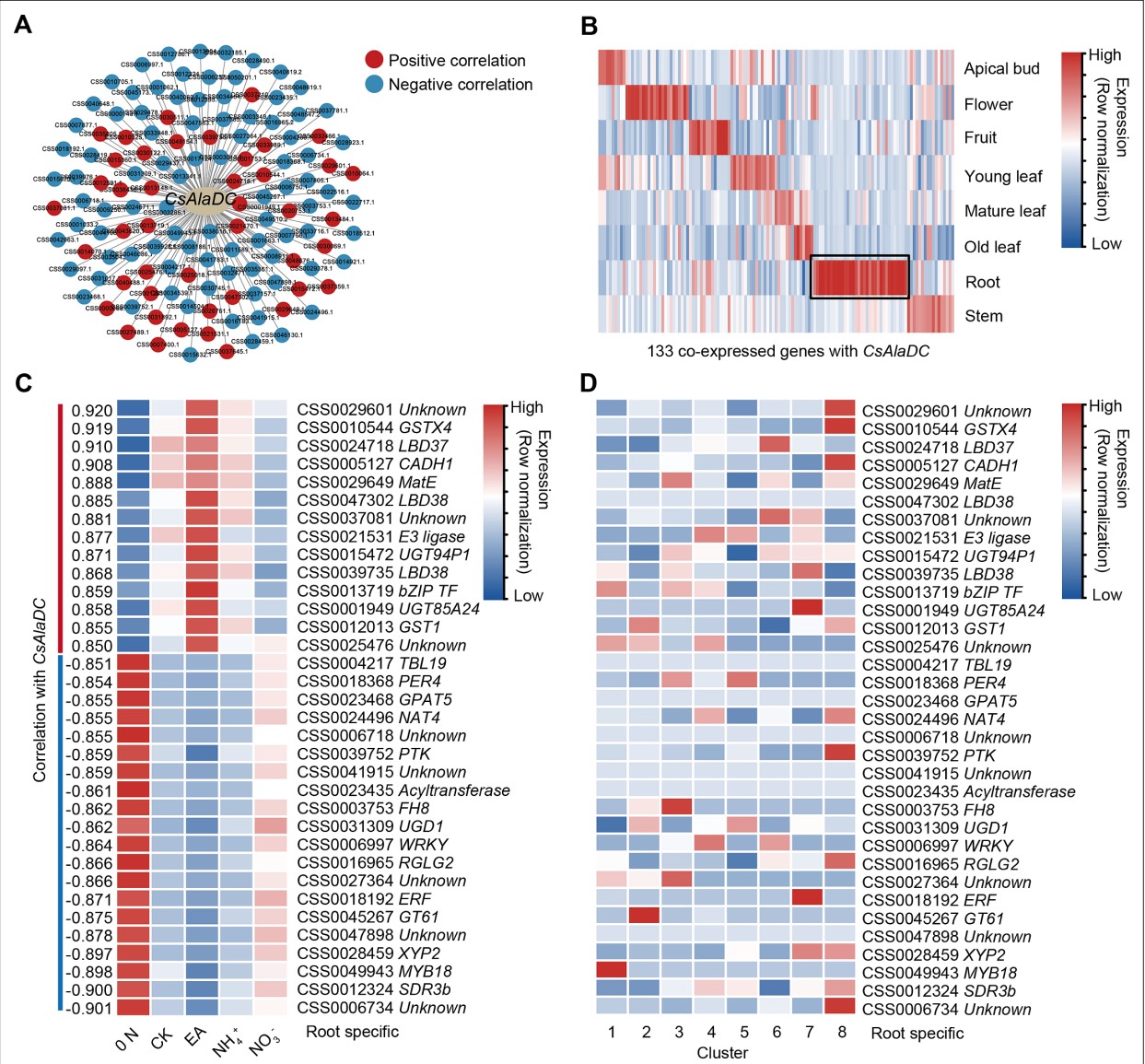

**Figure 6.** A gene co-expression network predicted key regulators of *CsAlaDC*. (**A**) Using bulk RNA-seq data, a gene co-expression network was generated for *CsAlaDC* (|correlation coefficient| ≥ 0.85 and p-value ≤0.05). Red and blue dots represent differential genes positively and negatively correlated with *CsAlaDC*, respectively. Bulk RNA-seq data are from different types of N treatment of tea plant roots (*Yang et al., 2020*). (**B**) Heatmap showing expression of co-expressed genes with *CsAlaDC* in eight tea plant tissues. The black arrows denote root-specific co-expressed (RSCE) genes. (**C**) Heatmap showing the expression pattern for RSCE genes under different types of N treatment (0 N, CK, EA, NH$_4^+$ and NO$_3^-$). Correlation between RSCE genes and *CsAlaDC* is shown on the heatmap at left. (**D**) Heatmap showing the expression level of RSCE genes in each cell cluster. (**B–D**) Row normalization of gene expression is according to the 'normalized' mode.

The expression patterns of these *CsAlaDC* co-expressing genes were assessed in eight tissues, based on comprehensive tea plant transcriptome data (*Wei et al., 2018*; *Xia et al., 2019*; *Figure 6B*). Given that *CsAlaDC* expression is root-specific, we chose 34 root-specific co-expressed genes (RSCGs) for further analysis (*Figure 6B*). Expression analyses of these RSCGs showed that the genes which were positively correlated with *CsAlaDC* expression were induced by N, especially by EA, and the negatively correlated genes were induced by a 0 N stress treatment (*Figure 6C*). Surprisingly, the results of the scRNA-seq analysis indicated that most of these genes, which were highly correlated with *CsAlaDC*, were not highly expressed in Cluster 1 (xylem cell) (*Figure 6D*).

Within these 34 root-specific genes, there were 7 genes potentially encoding for TFs LBD37, LBD38, bZIP, WRKY, AP2, and MYB18 (*Figure 6C*). Among them, *CsLBD37* (CSS0024718) had the most

positive correlation with *CsAlaDC*, and was induced by N (CK, EA, and $NH_4^+$) and was not expressed highly in Cluster 1 (*Figure 6C, D*). *Du et al., 2021* predicted that *CsLBD38*, named *CsLBD37* in this study, is a highly credible TF that regulates theanine synthesis and is in an evolution node of theanine-associated regulatory module in *Camellia* and Non-*Camellia* species. Therefore, CsLBD37 probably regulates *CsAlaDC* expression in tea plants, but its cell-type expression pattern is different from that of *CsAlaDC*.

To further explore this situation, we first confirmed the root-specific expression for *CsLBD37* (*Figure 7A, B*). Next, we performed *in situ* RT-PCR assays, which indicated that the *CsLBD37* expression pattern did not exactly reflect that observed for *CsAlaDC*. *CsLBD37* was expressed in the cambium; however, it was also expressed in the pericycle, whereas *CsAlaDC* was barely expressed in these cells (*Figures 5C and 7C, D*). As expected, CsLBD37 was localized in the nucleus (*Figure 7E*), and furthermore, yeast one-hybrid (Y1H) assays, combined with *CsAlaDC* promoter-*LUC* assays conducted in tobacco leaves, were consistent with CsLBD37 binding directly to and repressing the activity of the *CsAlaDC* promoter (*Figure 7F, H*). We further conducted an electrophoretic mobility shift assay (EMSA) with recombinant CsLBD37 protein and showed that it bound to the proximal region (−1734 to −1695 bp) of the *CsAlaDC* promoter (*Figure 7G*, *Figure 7—figure supplement 1*).

To test whether *CsLBD37* was able to inhibit *CsAlaDC* expression, *in vivo*, we developed transiently *CsLBD37*-silenced or overexpression tea seedlings, by antisense oligonucleotide (asODN) interference (*Figure 7I*) and generation of transgenic hairy roots, respectively (*Figure 7K*). *CsLBD37* expression was down-regulated and *CsAlaDC* was up-regulated in these asODN-treated roots (*Figure 7I*), whereas EA and theanine levels were increased (*Figure 7J*). In contrast, in *CsLBD37* transgenic hairy roots, *CsLBD37* was up-regulated and *CsAlaDC* was down-regulated (*Figure 7K*), whereas EA and theanine levels were decreased (*Figure 7L*). These findings offered support for the hypothesis that *CsLBD37* regulates *CsAlaDC* expression to modulate EA and theanine synthesis. *CsLBD37* probably acts as a 'brake' to maintain the expression level of *CsAlaDC* within a certain range, in response to high levels of N. This may be why *CsLBD37* is a repressor but is positively correlated with *CsAlaDC*.

Some studies have reported that the lateral organ boundaries domain (LBD) family of TFs can regulate both plant secondary metabolism and root development (*Goh et al., 2012*; *Okushima et al., 2007*; *Ye et al., 2021*). Lateral root founder cells originate from the pericycle, located opposite to the xylem poles (*Péret et al., 2009*). We noted that *CsLBD37* was also expressed in the tea plant root pericycle (*Figure 7C*). Therefore, we overexpressed *CsLBD37* in *A. thaliana* (*Figure 7—figure supplement 2A, B*) and observed that the number of lateral roots, in these overexpression lines, was significantly less than in the wild-type (WT) plants (*Figure 7—figure supplement 2B, C*). We further observed that *CsLBD37* overexpression repressed the initial steps of lateral root development (*Figure 7—figure supplement 2D–G*). These results implied CsLBD37 plays a role in regulates lateral root development in tea plant. Moreover, *CsLBD37* expression was induced by exogenous theanine treatment and this result underscores once again the importance of CsLBD37 in co-regulating theanine biosynthesis and root development (*Figure 7—figure supplement 2H*).

Collectively, our findings support a model in which transcriptional regulation of theanine biosynthesis does not require high expression of both TFs and target genes in the same cell cluster, but only requires partial co-expression. We also established that *CsLBD37* co-regulates both theanine biosynthesis and lateral root development in tea plants under N-rich conditions.

## Discussion

Application of single-cell transcriptome sequencing to plant studies has afforded progress in understanding root development in model plants, such as *A. thaliana* and rice (*Ryu et al., 2019*; *Zhang et al., 2021a*). In this regard, important aspects of secondary and specialized metabolism occur in the plant root system and are important for plant growth and adaption to environmental challenges. These metabolic products are also essential for food and beverage quality, and pharmaceutical products. *Bird et al., 2003* reported that biosynthesis of the narcotic analgesics, morphine and codeine, is localized to sieve elements in the opium poppy, which extends the function of sieve elements beyond transport, and into the realm of metabolism. However, significantly less information is currently available regarding both the cell heterogeneity of these secondary metabolic pathways and the underlying regulatory events that occur in the plant root system, especially in woody plants. The information regarding gene expression, in different type cells, is important with respect to establishing insights

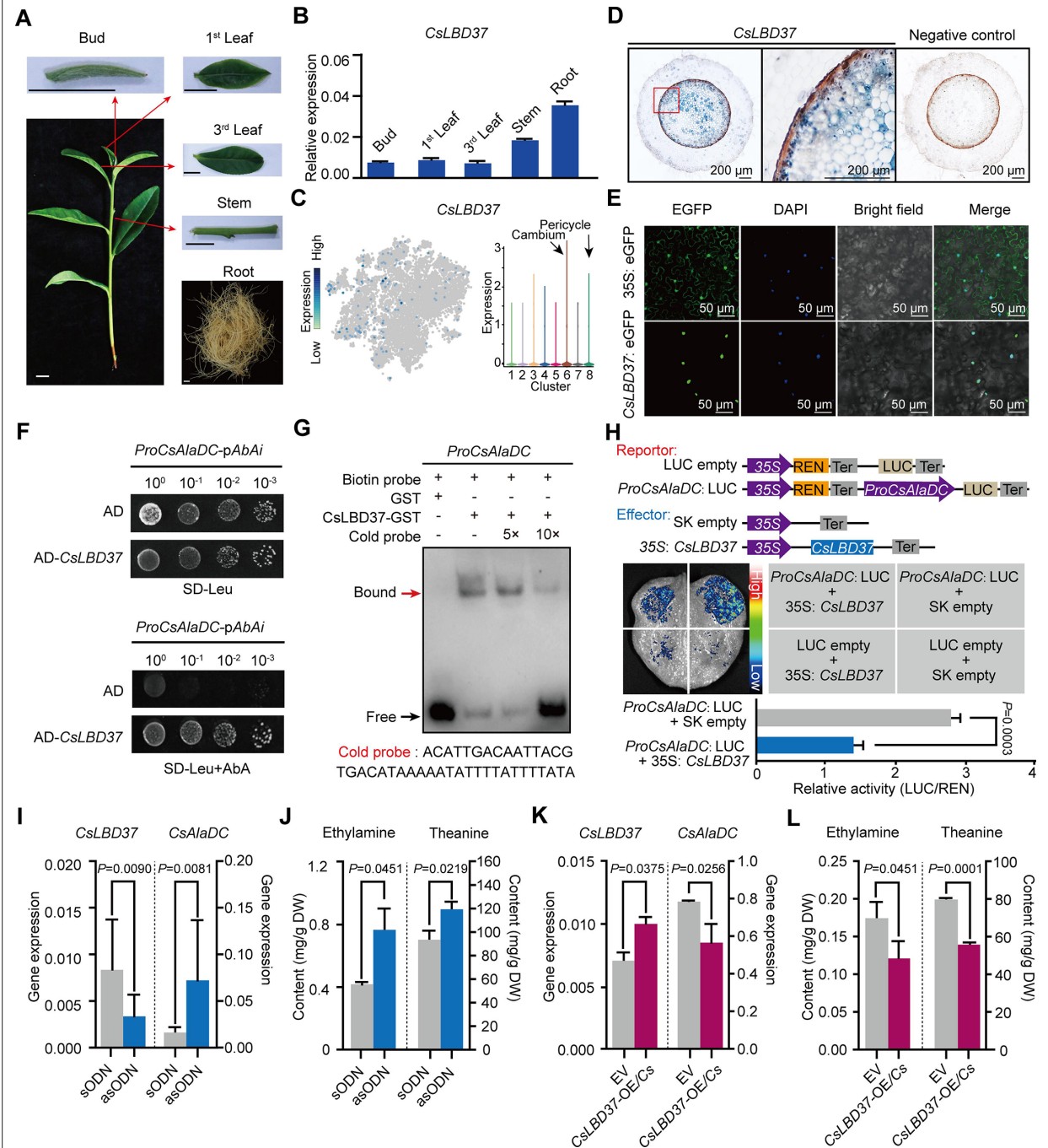

**Figure 7.** Theanine synthesis regulated by the transcription factor CsLBD37. (**A**) Schematic representation of the tea plant tissues used for gene detection. (**B**) Relative expression of *CsLBD37* in different tea plants tissues. (**C**) The *t*-distributed stochastic neighborhood embedding (*t*-SNE) visualization graph shows expression levels of *CsLBD37* in various cell clusters. (**D**) Cell specificity of *CsLBD37* expression in tea plant roots; blue signal represents location of gene expression. Scale bar = 200 μm. (**E**) Subcellular localization of CsLBD37; green fluorescence signal represents protein localization. Scale bar=50 μm. (**F**) The binding of CsLBD37 to the CsAlaDC promoter, tested by yeast one-hybrid (Y1H) assay. (**G**) Electrophoretic mobility shift assay (EMSA) showing the association of CsLBD37 with the *CsAlaDC* promoter. The red arrow points to the binding position. (**H**) A schematic of the effector and reportor constructs for transcriptional activity of the *CsAlaDC* promoter (top). LCI assays in tobacco leaves show that the *CsAlaDC* promoter can interact with CsLBD37. The color bar indicates the range of luminescence intensity (mid). The LUC/REN values represent relative activity of CsLBD37 on *CsAlaDC* expression. Data represent the means ± SD of three biological replicates (n=3). Significant difference was evaluated by two-tailed Student's *t*-test analysis (bottom). (**I–L**) *CsLBD37* gene silencing and overexpression in tea plant roots. Data represent the means ± SD of three biological replicates (n=3). Significant difference was determined by Welch's *t*-test. The expression level of *CsLBD37* and *CsAlaDC* in gene silenced (**I**) and

*Figure 7 continued on next page*

*Figure 7 continued*

overexpression plants (**K**). Ethylamine and theanine contents in gene silenced (**J**) and overexpression plants (**L**). sODN, sense oligo nucleotides; asODN, antisense oligonucleotides, silenced *CsLBD37* in tea plant root by asODN methods; EV, empty vector; *CsLBD37*-OE/*Cs*, *CsLBD37* overexpression in transgenic hairy root system.

The online version of this article includes the following source data and figure supplement(s) for figure 7:

**Source data 1.** PDF file containing original images of gels for *Figure 7G*, indicating the relevant bands and treatments.

**Source data 2.** Original files for images of gels displayed in *Figure 7G*.

**Figure supplement 1.** Protein purification of *CsLBD37*.

**Figure supplement 1—source data 1.** PDF file containing original images of gels for *Figure 7—figure supplement 1*, indicating the relevant bands and treatments.

**Figure supplement 1—source data 2.** Original files for images of gels displayed in *Figure 7—figure supplement 1*.

**Figure supplement 2.** Overexpression of *CsLBD37* in *Arabidopsis* inhibited lateral root development.

into both the mechanisms underlying these secondary metabolites and the regulatory events that control their syntheses. In this study, we performed a scRNA-seq-based study of tea plant roots, as an example, to elucidate the biosynthesis and regulatory processes for theanine, a root-specific secondary metabolite. In this process, we established the first map of tea plant roots comprised of eight cell clusters (*Figures 2A and 3D*).

## Cell cluster annotation of non-transgenic plants

Cell cluster annotation is a critical step for scRNA-seq. In this regard, we employed candidate marker genes to annotate cell clusters, as in previous studies (*Ryu et al., 2019*; *Zhang et al., 2021a*). In some non-model plants, including tea, transgenic technologies are not currently available and, hence, we used *in situ* RNA hybridization to establish the location(s) for gene expression. In some studies, isolation of different cell types was combined with quantitative RT-PCR (qRT-PCR) to detect cell-type marker gene expression (*Wang et al., 2022*). However, this approach has two limitations in that it cannot display the gene location directly and has only low resolution.

After numerous trials, we were able to optimize *in situ* RT-PCR assays (detailed in the Methods), which enabled a cell-specific characterization of gene expression in tea plant root cells, prior to establishing a genetic transformation system for tea. However, we note the challenge associated with weak calling of homologous marker genes, which may reflect the differences between tea and *A. thaliana*, along with differences in sampling sites, but this only slightly impacted our findings, as in the annotation of cell clusters, *in situ* RT-PCR was employed to identify expression patterns for weakly expressed tea plant root genes, such as *CsAAP1*, *CsLHW*, *CsAMT1.1*, and *CsNRT1.1*. Importantly, the results of *in situ* RT-PCR and scRNA-seq analysis were generally consistent.

## Nitrogen metabolism and transport of tea plant root at the single-cell level

N is one of the most critical mineral nutrients essential for crop growth and yield performance and provides precursors for the biosynthesis of secondary metabolites. Roots acquire $NO_3^-$-N and $NH_4^+$-N from the soil (*Xu et al., 2012*), with CsNRTs (CsNRT2.5 and CsNRT3.2) and CsAMTs (CsAMT3.1) acting in N uptake and transport in tea plant roots (*Zhang et al., 2021b*; *Zhang et al., 2020*). Our current findings provided further insight into the cellular locations where these transporters function. We show that many *CsNRTs* and *CsAMTs* are in the cortex, endodermal and stelar cells, whereas the N assimilation genes, *CsNRs* and *CsNiRs*, were primarily located within the stelar cells (*Figure 4B*). Our data support a model in which these CsNRTs and CsAMTs function in N uptake, into the cytoplasm of cortical and endodermal cells, but also play an important role in $NO_3^-$ and $NH_4^+$ loading into the cells within the stele for N assimilation and amino acids synthesis.

The Glu pathway is the most abundant and active pathway for amino acid biosynthesis in all root cell types. However, our findings indicated that stelar cells and the SCN have a higher capacity for amino acid synthesis, compared with epidermal, cortical, and endodermal cells (*Figure 4C*). For 20 amino acids, their rates of synthesis appeared to varying in different tea plant root cells; a similar situation was recently reported in *A. thaliana* leaves (*Kim et al., 2021*), but has not yet been reported

in the root systems of other plants. These findings deepen our understanding of N metabolism and transport in plant roots.

Numerous studies have established that the processes underlying plant N sensing, uptake, transport, and utilization are critical for the next novel green revolution (*Liu et al., 2022c*). In this regard, high-throughput scRNA-seq studies should offer greater insights into the cell-type specificity of many N-associated genes, which could help to select cell-type-specific genes more efficiently for future designed breeding programs.

## Multicellular compartmentation of theanine metabolism and transport

Compared with N metabolism in other plant species, theanine metabolism is a special component of N metabolism in tea plants (*Xu et al., 2012*; *Zhao et al., 2020*). Tea plants prefer $NH_4^+$ and synthesize large amounts of non-toxic theanine to decrease toxicity associated with high $NH_4^+$ levels (*Lin et al., 2022*). In the theanine biosynthesis pathway, and consistent with the findings of *She, 2022*, our study established that *CsTSI* is expressed primarily in the pericycle cells (Cluster 8). We further clarified that *CsAlaDC* is generally expressed in the vasculature (Clusters 1). This differential enrichment of *CsTSI* and *CsAlaDC* expression suggests that EA and theanine are synthesized in different cell types (*Figure 5*), presumably reflecting multicellular compartmentation (*Figure 8A*).

It is well known that theanine is the most abundant amino acid in the tea plant, and that theanine synthesis does not always occur, but is regulated to maintain growth and development in balance or adapted to environmental conditions. Indeed, the observed multicellular compartmentation may contribute to the regulation of theanine biosynthesis through precursor syntheses and transport within different cells. Thus, we propose a model in which multicellular compartmentation of theanine and its precursors functions like an innate design for regulation of secondary metabolism. This regulation model may be a way to avoid excessive or inhibition of theanine synthesis (*Figure 8A*). Compartmentalization of theanine biosynthesis may effectively prevent interference between secondary metabolic pathways. Similar mechanisms may also have evolved in other plants. We also cannot rule out that synthesis of theanine or EA can occur in the same cell type; after all, *CsTSI* and *CsAlaDC* are two of the most highly expressed genes in tea plant roots and their cell-type-specific expression partially overlapped (*Yang et al., 2020*; *Figure 4F*). In addition, the spatial distribution of theanine in the roots can be affected by transport processes. So, it is likely that the cell types in which theanine is distributed do not exactly correspond to the cell types in which theanine is synthesized.

On the other hand, multicellular compartmentation of theanine and its precursors EA implied EA transport from its location of biosynthetic into theanine biosynthetic cells. EA is a colorless gas, and hence it may be transported by way of simple diffusion, or via a gas channel. Perhaps aquaporins (AQPs), a class of channel proteins that transport water, gas, and nutrients (*Hachez and Chaumont, 2010*), may be involved. At present, the mechanism of EA transport remains unknown, and the involvement of cell-type-specific AQPs would be worthy to explore in the future.

Previously, we identified theanine transporters acting in theanine root-to-shoot transport and theanine storage in the vacuole of tea plant root cells (*Dong et al., 2020*; *Feng et al., 2021*). In the current study, we elucidated the cell heterogeneity of theanine transporters (*Figures 5B and 8B*), and among them *CsAAP1* is located primarily in cortical and endodermal cells (*Figures 2E and 5B*), and *CsCAT2* is co-expressed with *CsTSI* in the same cell type. This nicely reflects that CsAAP1 and CsCAT2 mediate in the physiological function of theanine transport and storage (*Dong et al., 2020*; *Feng et al., 2021*). However, these theanine transporters are all importers (*Dong et al., 2020*; *Feng et al., 2021*). The exporter(s) mediating in theanine transport from the xylem parenchyma for loading into the xylem transpiration stream, within the roots, along with other theanine export processes in the shoot, remain to be identified (*Figure 8B*). Combining the cell-type information, provided by scRNA-seq data, may assist in the identification of these specific-cell-located theanine exporter(s), and may also provide information for identifying amino acid exporters in other plant species. At present, all the exporters identified are bidirectional transporters, such as UMAMITs (*Besnard et al., 2016*). Therefore, specific-cell-located transporters, with bidirectional transport functions, could be potential theanine exporters.

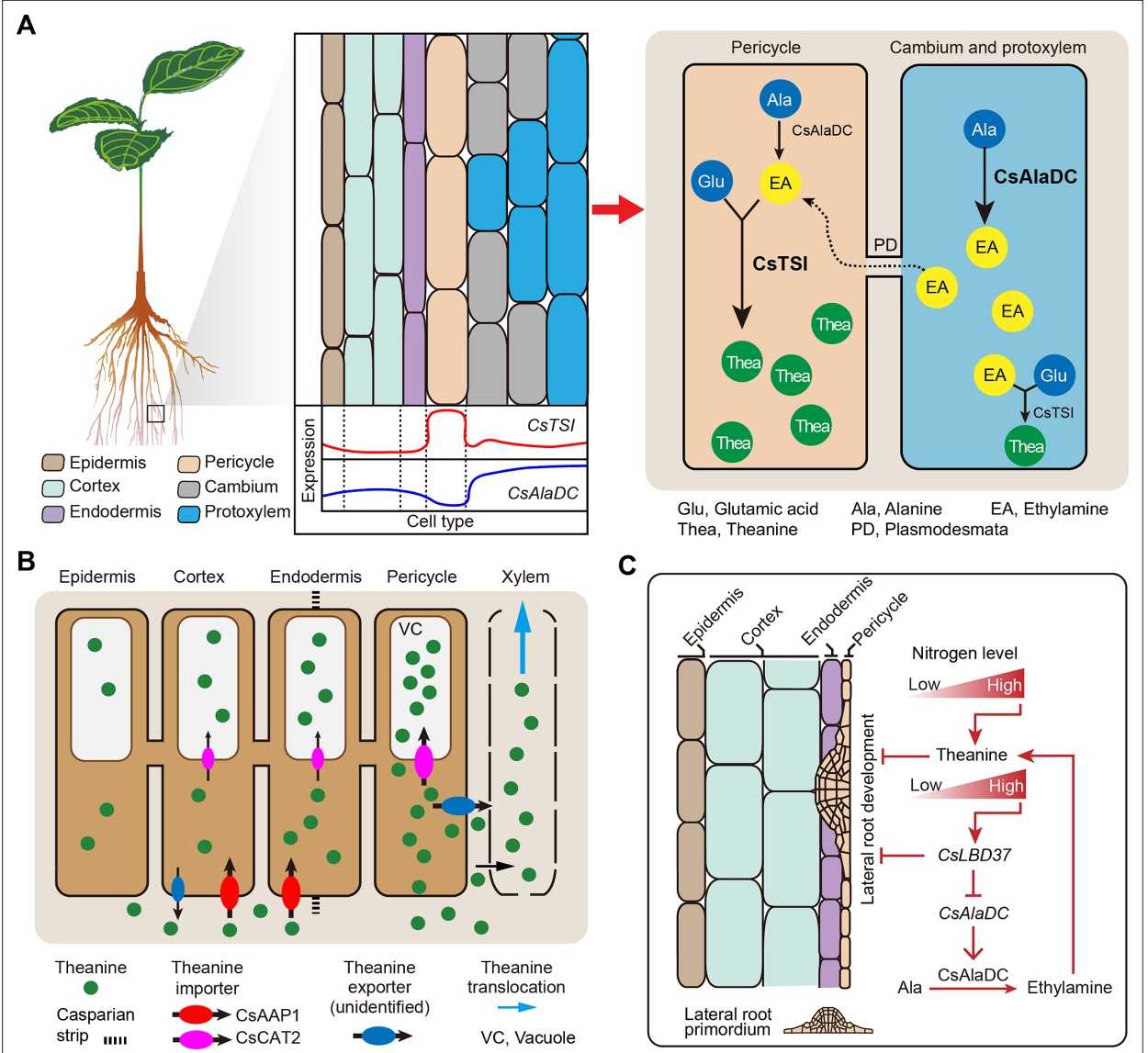

**Figure 8.** Proposed model for theanine biosynthesis, transport, and regulation at single-cell resolution. (**A**) Multicellular compartmentation of theanine biosynthesis. The high expression of *CsTSI* in pericycle cells suggests theanine is mainly synthesized in pericycle cells. *CsAlaDC* is highly expressed in cambium and protoxylem cells, suggesting that ethylamine (EA) is mainly synthesized in these cells. Thus, EA may move from cambium and protoxylem cells into pericycle cells, where it is a substrate for theanine biosynthesis. (**B**) In tea plant roots, CsAAP1 likely mediates in theanine retrieval from apoplast for its transport into cortical/endodermal cells. CsCAT2 imports theanine into vacuoles, especially in pericycle cells, which are the main cells for theanine biosynthesis. Theanine is exported, by an unidentified theanine exporter, from pericycle cells into the apoplastic pathway for entry into the xylem. (**C**) CsLBD37 co-regulates theanine biosynthesis and lateral root development. Increasing levels of nitrogen (N) promote theanine biosynthesis and accumulation. High accumulation of theanine and N induces *CsLBD37* expression. CsLBD37 inhibits *CsAlaDC* expression to reduce ethylamine synthesis, which finetunes theanine biosynthesis in feedback loop. Besides, *CsLBD37* is expressed in pericycle cells and also inhibits lateral root development. It is reported that apoplastic theanine can negatively regulate lateral root development, under high nitrogen levels (***Chen et al., 2023***). Therefore, CsLBD37 may also be involved in theanine-regulated lateral root development.

## The regulation of theanine biosynthesis at the single-cell level

Interestingly, genes highly correlated with *CsTSI* or *CsAlaDC* were identified, including some confirmed TFs (*CsMYB6*, *CsMYB40*, and *CsHHO3*) (***Guo et al., 2022***; ***Zhang et al., 2021c***), which were not always highly co-expressed in the same cell type. The cell heterogeneity of regulatory genes for *CsTSI* or *CsAlaDC* expression may reflect the 'accelerator'and 'brake' regulatory patterns that could act to fine-regulate theanine biosynthesis in specific cells (***Guo et al., 2022***). Another possibility is that these theanine-associated genes are promiscuous, having many target genes and regulate multiple

biological processes in tea plants. In the model plant *Arabidopsis*, AtARF7 and AtARF19 regulate *AtLBD16* and *AtLBD29*, thereby participating in the regulation root development (*Okushima et al., 2007*; *Figure 5—figure supplement 1B, C*). Meanwhile, AtARF7 and AtARF19 acted in *AtPHR1*, AtPHR1 activated *AtF3'H* and *AtRALF23* to co-regulate flavonoid metabolism and plant immunity (*Liu et al., 2022b*; *Tang et al., 2022*; *Figure 5—figure supplement 1B, C*). The highest expressed cell types of these regulators and target genes were different. Thus, TFs that regulate the theanine biosynthesis pathway may not always be co-express highly with their target genes, in the same cell type. Indeed, this point was confirmed in a subsequent study on CsLBD37 regulation of *CsAlaDC* expression.

Of course, TFs also will regulate their target genes and are highly co-expressed in the same cell type. As an example, AtMYB12 activated *AtCHS* and they were both highly expressed in cortical cells (*Stracke et al., 2007*; *Figure 5—figure supplement 1B, C*). However, members of the MYB–bHLH–WD40 (MBW) complex encoded genes were expressed in different cell clusters (*Xu et al., 2015*; *Figure 5—figure supplement 1*). Also, the processes of gene regulation may be dynamically changing. Some TFs, such as HY5, SHR, and TMO, can alter their expression in different cell types, due to their mobile characteristics or changes in cell differentiation and external conditions. (*Ortiz-Ramírez et al., 2021*; *Chen et al., 2024*; *Mankotia et al., 2024*). In short, gene regulation is complicated at the single-cell level. This complexity includes the underlying nature of the biological processes involved, along with the complexity and dynamics of their regulatory mechanisms. In this regard, ScRNA-seq better enabled us to identify more details about gene regulation that were not previously noted.

To date, some identified TFs have been shown to play a role in the fine-regulation of *CsTSI* and *CsAlaDC* expression; however, the TFs that impart this *CsTSI* and *CsAlaDC* root specificity and high expression remain to be identified. Recently, it was established that, in roots, hypomethylation in the promoters of *CsTSI* and *CsAlaDC* provided an epigenetic basis for their high expression (*Kong et al., 2023*). Although hypomethylation in the promoters provides accessibility for transcription, activating TFs is still required for their high expression. In this regard, the current cell-type-specific expression database offers a valuable resource to identify such unknown TF activators. For example, root- and Cluster 1-specific CsMYB18 could well be an activator of *CsAlaDC* expression (*Figure 7B–D*).

## Crosstalk between theanine metabolism and root development

Recently, some TFs regulating *CsTSI* or *CsAlaDC* were identified and characterized and indicted that theanine regulates lateral root development (*Chen et al., 2023*; *Guo et al., 2022*; *Zhang et al., 2021c*); however, the molecular link between these processes was not elucidated. Previous studies have shown that several LBD family members regulate proanthocyanidin (PA) metabolism and are involved in root development in response to N in *A. thaliana* (*Rubin et al., 2009*). In tea plant roots, theanine, PA metabolism and root development are also regulated by N (*Wang et al., 2021*). In addition, *CsLBD37* was also N-inducible and was one of the most positively correlated TF encoding genes with *CsAlaDC* (*Figure 6C*). This information provided important insights into the roles for *CsLBD37* in theanine biosynthesis and root development. In this study, we identified CsLBD37 as a repressor of *CsAlaDC* transcription and root development (*Figures 7 and 8C*, *Figure 7—figure supplement 2*). CsLBD37 may act in a similar manner to CsHHO3, namely through being a repressor of *CsAlaDC* transcription to 'brake' theanine biosynthesis under high N availability (*Guo et al., 2022*). Here, we also revealed that *CsLBD37* probably inhibits lateral root development, at the root initial stage (*Figure 7—figure supplement 2D–G*), but the underlying mechanism remains unknown. *CsLBD37* expression could be induced by exogenous theanine treatment in tea seedlings (*Figure 7—figure supplement 2H*). This implied that CsLBD37 may also be involved in a process by which theanine regulates lateral root development. Besides, it has been reported that LBD family TFs were regulated by, or interacted with, regulators of hormone pathways (e.g., ARF family TFs) to regulate the process of root morphogenesis (*Goh et al., 2012*; *Ye et al., 2021*). Based on these findings, we speculated that CsLBD37 is likely regulated by, or interacts with, other proteins to form a complex to regulate root development or theanine biosynthesis. Then, is *CsLBD37* involved in regulating root development through the hormones and ROS homeostasis pathways as well as theanine? This question is worthy of future investigation.

In summary, we generated a transcriptional map of tea plant roots, at the single-cell level, which provided cell-type-specific information on the expression of some 40,000 genes. Moreover, these

datasets provide an important resource for future studies on secondary metabolism within tea plant roots, as well as other plant root systems. Recently, spatial transcriptomics and metabolomics techniques were applied for studies in both animals and plants (*La Manno et al., 2021*; *Liu et al., 2022a*). An integration of these techniques with scRNA-seq would afford a powerful approach to further explore cell-specific secondary metabolism, at higher resolution, in various crop root systems.

## Materials and methods
### Plant materials and growth conditions
Tea plants (*C. sinensis* var. *sinensis* cv. *Shuchazao*) seeds were soaked in water for 7 days until their seed coats burst: water was changed daily. Tea seeds were then cultured in vermiculite and watered (twice weekly) for 30 days, until radicle emergence. These tea seeds were then transplanted, singly, to new vermiculite, and watered (three times weekly) until the two-leaf stage, when they were transferred to nutrient solution for further growth; these hydroponic conditions and nutrient solution were as described previously (*Konishi et al., 1985*).

### Preparation of root samples for scRNA-seq
The root tip regions (~2 cm in length from root tip) were cut, longitudinally, into three to four small strips and then processed for protoplast isolation. Here, the root strips were transferred immediately and gently into the prepared enzyme solution (0.1 mol l$^{-1}$ KCl, 0.08 mol l$^{-1}$ MES (2-Morpholinoethanesulphonic acid), 0.02 mol l$^{-1}$ CaCl$_2$, 1.5% cellulase R10 (Yakult, Japan), 1% Pectolyase (Yakult, Japan), 0.4 mol l$^{-1}$ mannitol, and 0.1% BSA (bovine serum albumin)), and digested at 25°C, with shaking (40 rpm), in the dark for the indicated period. An equal volume of washing buffer (WB) solution (0.1 mol l$^{-1}$ KCl, 0.08 mol l$^{-1}$ MES, 0.4 mol l$^{-1}$ mannitol and 0.1% BSA) was added in the enzyme solution and the mixture was then filtered using nylon mesh (200 mesh/inch). The flow-through was centrifuged at 200 × *g* for 2 min to pellet protoplasts. To evaluate protoplast yield, under different isolation conditions, protoplasts were resuspended with suitable WB solution and counted with a hemocytometer. Protoplast viability was determined by trypan blue staining, with the ratio of viable cells being more than 85%. The concentration of protoplasts was adjusted to 700–1200 cells/μl and cells were then processed with the 10× Genomics Single Cell Protocol (CG00052, RevC).

### Construction and sequencing of the scRNA-seq library
Approx. 12,593 counted cells were loaded onto a Single Cell Gchip. The libraries were constructed using 10× Genomics Chromium Next GEM Single Cell 3′ Reagent Kits v3.1 (1000268). In brief, cell suspensions in a chip were loaded on a Chromium Controller (10× Genomics, Pleasanton, CA) to generate single-cell GEMs (gel beads in emulsion). The scRNA-seq libraries were then prepared, according to the manufacturer's protocol. Qualitative analysis of the DNA library was performed on an Agilent 2100 Bioanalyzer. The concentration of DNA library was measured by Qubit (Invitrogen). Libraries were sequenced by an Illumina NovaSeq sequencer (Genergy Biotechnology Shanghai) and the raw scRNA-seq dataset comprised Read1, Read2, and i7 index read. The 26 bp read length of Read1 contained the sequence of the 16 bp 10xBarcode and 10 bp UMI (unique molecular identifiers). The 98 bp read length of Read2 was the sequence of the cDNA fragment. In total, 332,593,050 reads were obtained.

### Pre-processing of raw scRNA-seq data
The raw files were analyzed by Cell Ranger 5.0.0 (10× Genomics). The tea plant genome and GTF annotation files, which excluded the organelle genomes, were downloaded from the Tea Plants Information Archive (TPIA) website (http://tpia.teaplants.cn) (*Xia et al., 2019*). Running 'cellranger mkref' with '–genome, –fasta and –genes' inputs was employed to build the reference. Then 'cellranger count' with '–id, –transcriptome, –fastqs, –sample' was run. Some 79.8% of the reads in all of the samples were aligned to the tea plant genome by the aligner STAR (v.2.5.1b). Mean reads per cell were 26,411. Median genes per cell were in the range of 680.

## Data integration, clustering, and annotation

Downstream analyses were mainly performed with the Seurat package (v.3.1.2), as previously described (*Butler et al., 2018*). To remove low quality cells and likely multiplet captures, which is a major concern in microdroplet-based experiments, we applied a criteria to filter out cells with UMI/gene numbers out of the limit of the mean value ± two fold of the standard deviations, assuming a Guassian distribution of the UMI/gene number for each cell. Additionally, we applied DoubletFinder package (version 2.0.2) potential doublet identification (*McGinnis et al., 2019*).

Library size normalization was performed with the 'NormalizeData' function (LogNormalize method, scaling factor of 10,000). We then detected variable genes with the 'FindVariableFeatures' function (vst method, 4000 features), scaled data with 'ScaleData' function, performed PCA analysis with 'RunPCA' function (14 principal components), determined statistical significance of PCA scores by 'JackStraw' function, constructed the SNN graph, clustered cells based on Louvain ('FindNeighbors' and 'FindClusters'), and visualized data with non-linear dimensional reduction algorithms ('RunTSNE'). The cluster-enriched genes were computed with the FindAllMarkers function in Seurat, using the following parameters: a bimod test; above 1 − fold difference (logfc. Threshold ≥0) between the two groups of cells; test genes that having a minimum fraction was at least 0.25.

## Pseudo-time analysis

Monocle2 (V2.9.0) package was used to infer the trajectory of cell differentiation (*Trapnell et al., 2014*), and the specific steps were as follows: First, the import CDS function of Monocle2 package was used to transform the Seurat object into the Cell Data Set object. 0.01, next the reduce Dimension function was used for dimensionality reduction clustering, and finally, order Cells function was used to infer the differentiation trajectory.

## *In situ* RT-PCR

*In situ* RT-PCR was improved and performed based on previous methods (*Munns et al., 2012*). The experimental material was the primary root of a 2-month-old 'shuchazao' tea cultivar. The young root tissue was soaked in fresh FAA solution, vacuum-infiltrated for 15–20 min, then put at 4°C for 12 hr. The samples were then eluted, three times, for 10 min each, in a mixture solution (63% ethanol and 5% acetic acid), and twice for 5 min each in 1× PBS (Phosphate buffer saline). Root tissue was then embedded in 5% low-melting point agarose (agarose dissolved in 1× PBS) and the embedded blocks were sectioned using a Leica RM2255 microtome (Leica, Nussloch, Germany). The root tissue sections were separated from agarose by rinsing, twice, with Rnase-Free water, 3 µg/ml proteinase K was then added for 30 min at 25°C. A heat treatment, at 85°C for 2 min, was given to inactivate proteinase K, and samples were then washed, once, separately with 1× PBS and then Rnase-Free water for 5–10 min, followed by the addition of 1 U/µl Dnase I, at 37°C, for 20 min, or overnight. Liquid was then removed and 15 mM EDTA (Ethylene diamine tetraacetic acid, pH 8.0) was added, at 75°C for 10 min, followed by one to two washes with Rnase-Free water to remove excess liquid. Reverse transcription reactions were performed using the PrimeScript II 1st strand cDNA Synthesis Kit (TAKARA, Cat. No. 9750), solution was then removed, followed by a wash, once, with Rnase-Free water. Mixed tissue sections and reaction reagents: 2 µl 10× Tag DNA polymerase (NEB, Cat. No. M0267V), 0.4 µl Buffer Tag DNA polymerase (5 U/µl, NEB, Cat. No. B9004S), 1.6 µl dNTP (TAKARA, Cat. No. 4030), 0.32 µl DIG-11-dUTP (25 nmol, Sigma, Cat. No. 11093088910), 1 µl Forward Primer, 1 µl Reverse Primer, 0.6 µl $MgCl_2$ (1.5 mM, Thermo Fisher, Cat. No. R0971) and 13.08 µl $ddH_2O$. PCR program: 95°C for 30 s, 95°C for 30 s, 55°C for 50 s, 68°C for 45 s, 30–35 cycles, 68°C for 5 min 10°C. Washed twice with 1× PBS for 5 min each, and blocked with confining liquid (5% skim milk) for 30 min. Alkaline phosphatase antibody was diluted in confining liquid at 1:500, 50 µl and was added and allowed to stand for 1 hr. Wash twice for 15 min each time using 10× WB. Sample staining was performed using BM purple AP substrate, precipitating for 30 min or 1 hr, then washed twice with Rnase-Free water, and observed under a microscope (Carl Zeiss, Gottingen, Germany) and photographed. All primers used for *in situ* RT-PCR are listed in *Supplementary file 9*.

## Analysis of theanine and EA content

Theanine and EA were exacted and detected as previously described, with slight modification (*Yang et al., 2020*; *Cheng et al., 2017*). Theanine was analyzed by a Waters e2695 HPLC system equipped

with 2489 UV/Vis detector (Waters, USA). EA was analyzed by GC–MS detecting (Agilent, Santa Clara, USA).

## WGCNA analysis

The WGCNA package, in R software, was used in co-expression analyses to detect relative relationships among genes (*Langfelder and Horvath, 2008*). A weighted adjacency matrix was created, following unsupervised hierarchal clustering analysis of genes, as follows: (1) removal of outliner genes and samples by the variation efficient and hierarchical clustering analysis; (2) choosing of an applicable soft threshold (power = 12); (3) identification of co-expressed gene modules using the function blockwise Modules; (4) visualization of the co-expression network using Cytoscape3.9.0.

## GO analysis

Tbtools (v1.098769) software was used to analyze the GO enrichment of co-expression genes (*Chen et al., 2020*). GO annotation texts of tea plants were summarized from TPIA database (http://tpia.teaplants.cn) (*Xia et al., 2019*).

## Subcellular localization

Sequence information on *CsLBD37* was obtained from TPIA (http://tpia.teaplants.cn) (*Xia et al., 2019*). The open reading frame of *CsLBD37* within the entry vector pDONR207 was cloned into the destination binary vector, pK7WGF2.0, for subcellular localization studies. The plasmid, pK7WGF2.0-*CsLBD37*-GFP was transformed into *Agrobacterium tumefaciens* strain EHA105 to select for positive colonies for infiltration into *Nicotiana benthamiana*. After 48–72 hr post-infiltration, the GFP fluorescence was imaged with an Olympus FV1000 confocal microscope (Olympus, Tokyo, Japan) to reveal the subcellular localization of *CsLBD37*. GFP fluorescence signals were excited with a 488-nm laser, and the emitted light was recorded from 500 to 530 nm to display the subcellular localization of *CsLBD37*.

## Y1H assays

We inserted *CsAlaDC* promoter fragments (approx. 2000 bp) into the pAbAi vector to generate the Pro*CsAlaDC*-pAbAi reporter construct, which was then introduced into the Y1H gold strain. The transformed cells were grown on synthetic dropout medium (SD–Ura) (Coolaber, Beijing, China). The ORF of *CsLBD37* was inserted into vector pGADT7 AD to generate AD-CsLBD37, which was then introduced into the yeast strain with pro*CsAlaDC*-pAbAi. Transformants were grown on synthetic dropout medium (SD-Leu) with 200 ng ml$^{-1}$ Aureobasidin A (AbA) (Coolaber, Beijing, China) for 3 days at 30°C. pGADT7 AD empty vectors were used as controls.

## Luciferase activity assay

Approx. 2 kb *CsAlaDC* promoter fragments were inserted into the pGreenII 0800-LUC vector to generate the luciferase reporter construct, pro*CsAlaDC*-LUC. The *CsLBD37* ORF was inserted into the pGreenII 62-SK vector to generate the effector construct *35S-CsLBD37*. Empty vectors were used as negative controls. The above-described constructs were introduced into *A. tumefaciens* strain GV3101 with the pSoup plasmid, respectively. Then effector and reporter cell suspensions were mixed, in a 5:1 (vol/vol) ratio, and infiltrated into 4-week-old *N. benthamiana* leaves, followed by harvesting after 3 days. Sections of leaves were sprayed with 150 mg l$^{-1}$ D-luciferin, potassium salt (Sciencelight, Shanghai, China) and LUC signals were captured using a Tanon 5200 Multi charge-coupled device camera (Tanon, Shanghai, China). LUC and REN activities of other *N benthamiana* leaves were measured using a dual-luciferase assay kit (Yeasen Biotech, Shanghai, China) and a Spectra Max M2 (Molecular Devices, America) according to the manufacturer's instructions.

## Electrophoretic mobility shift assay

The *CsLBD37* ORF was inserted into the GST vector pGEX-4T-1 to generate GST-CsLBD37 construct, which was then transferred into *Escherichia coli* BL21 cells. The positive BL21 strains were induced by 0.5 mM isopropyl-beta-D-thiogalactopyranoside, at 16°C for 20 hr, and the GST-CsLBD37 recombinant proteins were purified by immobilized glutathione beads (TransGen Biotech, Beijing, China) (*Figure 7—figure supplement 1*). DNA probes were synthesized and labeled with biotin, at the 3'

end, by using the EMSA Probe Biotin Labeling Kit (Beyotime Biotech, Shanghai, China). The EMSA was performed using the instructions of the Chemiluminescent EMSA Kit (Beyotime Biotech, Shanghai, China).

## Antisense oligonucleotides assay

The antisense oligonucleotides assay was performed, as previously described (*Xie et al., 2014*), to silence the target gene *CsLBD37*. AsOND was designed and selected using the SOLIGO software (https://sfold.wadsworth.org/cgi-bin/soligo.pl). The sequence of the sODN and AsODN is listed in *Supplementary file 9*. To silence the genes, the primary roots of tea plant cultivar 'Shuchazao' were wounded and then incubated in 10 ml 50 µM asODN-*CsLBD37* solution for 6 and 12 hr. The solution with sense oligonucleotides (sOND) of the gene serves as the control. After treatment, the roots were collected for RNA isolation and metabolite detection.

## Overexpression of *CsLBD37* in a tea plant hairy root system

The *CsLBD37* ORF was inserted into pK7WGF2.0 driven by the *35S* promoter, using the gateway system to generate the pK7WGF2.0-CsLBD37 overexpression construct. The resultant vector was transformed into Ar.A4 Chemically Competent Cell (Weidi Biotech, Shanghai, China). The selected positive transformants were collected and resuspended in 1/2 Murashige and Skoog (MS) culture medium to an $OD_{600}$ = 0.6 to transform 3-month-old tea seedlings, as described previously (*Alagarsamy et al., 2018*).

## *Arabidopsis* transformation

The Gateway system was employed to generate the pB2GW7-CsLBD37 construct. *A. tumefaciens* strain GV3101, carrying pB2GW7-*CsLBD37*, was used for the genetic complementation of *Arabidopsis* wild-type (Col-0) plants. Transformants were selected on 0.25 ‰ Basta, and further verified by genomic PCR and qRT-PCR. The T3 progeny plants were used for these analyses. All primers are listed in *Supplementary file 9*.

## Theanine treatment of tea seedlings

Tea seedlings with a 5 cm primary root were cultured in 1/2 nutrient solution for 1 week, and then treated with 1/2 nutrient solution containing theanine for 2 weeks. The theanine concentration of the control group (CK) was 0 mM and that for the treatment group (Theanine) was 1 mM.

## RNA isolation and q-RT-PCR

RNA isolation was performed according to the manual of RNAprep Pure Plant Plus Kit (TIANGEN, Beijing, China), and cDNA synthesis was accomplished using the TransScript II One-Step gDNA Removal and cDNA Synthesis SuperMix kit (TransGen Biotech, Beijing, China). A QuantStudio 6 Flex Real-Time PCR System (Thermo, Singapore) was used to measure relative expression of genes, which were calculated based on the $2^{-\Delta Ct}$ method (*Schmittgen et al., 2000*), using *CsGAPDH* as an internal control. All primers used for qRT-PCR are listed in *Supplementary file 9*.

# Acknowledgements

This work was supported by the National Key R&D Program of China (2022YFF1003103, 2021YFD1601101), and grants from the National Natural Science Foundation of China (32072624), Anhui Provincial Major Science and Technology Project (202103b06020024), and Anhui Educational Committee Excellent Youth Talent Support project (gxyqZD2022018). We want to thank Oebiotech company provide 10× scRNA-seq service.

# Additional information

## Funding

| Funder | Grant reference number | Author |
|---|---|---|
| National Key R&D Program of China | 2022YFF1003103 | Xiaochun Wan |
| National Key R&D Program of China | 2021YFD1601101 | Zhaoliang Zhang |
| The National Natural Science Foundation of China | 32072624 | Zhaoliang Zhang |
| Anhui Provincial Major Science and Technology Project | 202103b06020024 | Tianyuan Yang |
| Anhui Educational Committee Excellent Youth Talent Support project | gxyqZD2022018 | Tianyuan Yang |

The funders had no role in study design, data collection, and interpretation, or the decision to submit the work for publication.

## Author contributions

Shijia Lin, Conceptualization, Data curation, Software, Formal analysis, Validation, Investigation, Visualization, Methodology, Writing - original draft, Project administration, Writing - review and editing; Yiwen Zhang, Data curation, Formal analysis, Validation, Investigation, Visualization, Methodology; Shupei Zhang, Yijie Wei, Mengxue Han, Yamei Deng, Jiayi Guo, Biying Zhu, Investigation; Tianyuan Yang, Funding acquisition, Guidance on experimental techniques; Enhua Xia, Guidance on bioinformatics analysis; Xiaochun Wan, Supervision, Funding acquisition, Project administration; William J Lucas, Offered suggestions on the study and made revisions to the writing of this manuscript; Zhaoliang Zhang, Supervision, Funding acquisition, Writing - original draft, Project administration, Writing - review and editing

## Author ORCIDs

Shijia Lin http://orcid.org/0009-0005-9762-0349
Biying Zhu https://orcid.org/0009-0000-9652-7787
Zhaoliang Zhang https://orcid.org/0000-0001-6615-1598

Reviewer #1 (Public review): https://doi.org/10.7554/eLife.95891.3.sa1
Reviewer #2 (Public review): https://doi.org/10.7554/eLife.95891.3.sa2
Reviewer #3 (Public review): https://doi.org/10.7554/eLife.95891.3.sa3
Author response https://doi.org/10.7554/eLife.95891.3.sa4

# Additional files

## Supplementary files

- MDAR checklist
- Supplementary file 1. QC of scRNA-seq.
- Supplementary file 2. Gene expression of scRNA-seq.
- Supplementary file 3. Top 10 marker gene of cell cluster.
- Supplementary file 4. Pseudotime heatmap gene module.
- Supplementary file 5. Nitrogen assimilation and amino acid synthesis.
- Supplementary file 6. Theanine synthesis and transport.
- Supplementary file 7. Gene expression of five nitrogen forms treatment.
- Supplementary file 8. Co-expression analysis of CsAlaDC and other genes.

• Supplementary file 9. Primer list of this study.

## Data availability

Sequencing data have been deposited in GEO under accession code GSE267845.

The following dataset was generated:

| Author(s) | Year | Dataset title | Dataset URL | Database and Identifier |
|---|---|---|---|---|
| Lin S, Zhang Y, Zhang S, Wei Y, Han M, Deng Y, Guo J, Zhu B, Yang T, Xia E, Wan X, Lucas WJ, Zhang Z | 2025 | Root-specific secondary metabolism at the single-cell level: a case study of theanine metabolism and regulation in the roots of tea plants (Camellia sinensis) | https://www.ncbi.nlm.nih.gov/geo/query/acc.cgi?acc=GSE267845 | NCBI Gene Expression Omnibus, GSE267845 |

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
