## [Editor Report · eLife assessment]

This study combines experimental and theoretical approaches to examine metabolites at the single-cell level in tea plants. The authors skilfully integrated various tools available for this type of research, and meticulously presented and illustrated every step of the survey. The overall quality of the work is **convincing**, and it represents an **important** contribution to our understanding of the compartmentalization of biosynthesis pathways.

---

## [Referee Report · Reviewer #1 (Public review)]

Summary:

The study used root tips from semi-hydroponic tea seedlings. The strategy followed sequential steps to draw partial conclusions.

Initially, protoplasts obtained from root tips were processed for scRNA-seq using the 10x Genomics platform. The sequencing data underwent pre-filtering at cell and gene levels, leading to 10,435 cells. These cells were then classified into eight clusters using t-SNE algorithms. The present study scrutinised cell typification through protein sequence similarity analysis of homologs of cell type marker genes. The analysis was conducted to ensure accuracy using validated genes from previous scRNA-seq studies and the model plant *Arabidopsis thaliana*. The cluster cell annotation was confirmed using *in situ* RT-PCR analyses. This methodology provided a comprehensive insight into the cellular differentiation of the sample under study. The identified clusters, spanning 1 to 8, have been accurately classified as xylem, epidermal, stem cell niche, cortex/endodermal, root cap, cambium, phloem, and pericycle cells.

Then, the authors performed a pseudo-time analysis to validate the cell cluster annotation by examining the differentiation pathways of the root cells. Lastly, they created a differentiation heatmap from the xylem and epidermal cells and identified the biological functions associated with the highly expressed genes.

Upon thoroughly analysing the scRNA-seq data, the researchers delved into the cell heterogeneity of nitrate and ammonium uptake, transport, and nitrogen assimilation into amino acids. The scRNA-seq data was validated by *in situ* RT-PCR. It allows the localisation of glutamate and alanine biosynthetic enzymes along the cell clusters and confirms that both constituent the primary amino acid metabolism in the root. Such investigation was deemed necessary due to the paramount importance of these processes in theanine biosynthesis since this molecule is synthesised from glutamate and alanine-derived ethylamine.

Afterwards, the authors analysed the cell-specific expression patterns of the theanine biosynthesis genes, combining the same molecular tools. They concluded that theanine biosynthesis is more enriched in cluster 8 "pericycle cells" than glutamate biosynthesis (Lines 271-272). However, the statement made in Line 250 states that the highest expression levels of genes responsible for glutamate biosynthesis were observed in Clusters 1, 3, 4, 6 and 8, leading to an unclear conclusion.

The regulation of theanine biosynthesis by the MYB transcription factor family is well-established. In particular, CsMYB6, a transcription factor expressed specifically in roots, has been found to promote theanine biosynthesis by binding to the promoter of the TSI gene responsible for theanine synthesis. However, their findings indicate that CsMYB6 expression is present in Cluster 3 (SCN), Cluster 6 (cambium cells), and Cluster 1 (xylem cells) but not in Cluster 8 (pericycle cells), which is known for its high expression of CsTSI. Similarly, their scRNA-seq data indicated that CsMYB40 and CsHHO3, which activate and repress CsAlaDC expression, respectively, did not show high expression in Cluster 1 (the cell cluster with high CsAlaDC expression). Based on these findings, the authors speculated that transcription factors and target genes are not necessarily always highly expressed in the same cells.

Lastly, the authors have discovered a novel transcription factor belonging to the Lateral Organ Boundaries Domain (LBD) family known as CsLBD37 that can co-regulate the synthesis of theanine and the development of lateral roots. The authors observed that CsLBD37 is located within the nucleus and can repress the CsAlaDC promoter's activity. To investigate this mechanism further, the authors conducted experiments to determine whether CsLBD37 can inhibit CsAlaDC expression in vivo. They achieved this by creating transiently CsLBD37-silenced or over-expression tea seedlings through antisense oligonucleotide interference and generation of transgenic hairy roots. Based on their findings, the authors theorise that CsLBD37 regulates CsAlaDC expression to modulate the synthesis of ethylamine and theanine in tea roots. Apologies for the inadvertent mistake concerning glutamate and glutamine.

Strength:

The manuscript showcases significant dedication and hard work, resulting in valuable insights that are fundamental for generating knowledge. The authors skillfully integrated various tools available for this type of study and meticulously presented and illustrated every step involved in the survey. The overall quality of the work is exceptional, and it would be a valuable addition to any academic or professional setting.

Weaknesses:

The authors have effectively addressed the feedback and revised the manuscript, presenting their debatable conclusions as speculative. Consequently, I find the manuscript's current form free of any apparent weaknesses.

---

## [Referee Report · Reviewer #2 (Public review)]

Summary:

In their manuscript, Lin et al. present a comprehensive single-cell analysis of tea plant roots. They measured the transcriptomes of 10,435 cells from tea plant root tips, leading to the identification and annotation of 8 distinct cell clusters using marker genes. Through this dataset, they delved into the cell-type-specific expression profiles of genes crucial for the biosynthesis, transport, and storage of theanine, revealing potential multicellular compartmentalization in theanine biosynthesis pathways. Furthermore, their findings highlight CsLBD37 as a novel transcription factor with dual regulatory roles in both theanine biosynthesis and lateral root development.

Strengths:

This manuscript provides the first single-cell dataset analysis of roots of the tea plants. It also enables detailed analysis of the specific expression patterns of the gene involved in theanine biosynthesis. Some of these gene expression patterns in roots were further validated through in-situ RT-PCR. Additionally, a novel TF gene CsLBD37's role in regulating theanine biosynthesis was identified through their analysis.

Weaknesses:

The revised manuscript has addressed the concerns raised during the initial review.

---

## [Referee Report · Reviewer #3 (Public review)]

Summary:

Lin et al., performed a scRNA-seq-based study of tea roots, as an example, to elucidate the biosynthesis and regulatory processes for theanine, a root-specific secondary metabolite, and established the first map of tea roots comprised of 8 cell clusters. Their findings contribute to deepening our understanding of the regulation of the synthesis of important flavor substances in tea plant roots. They have presented some innovative ideas.

Comment on revised version:

The reviewer has addressed all my concerns and I have no further comments.

---

## [Author Response]

The following is the authors’ response to the original reviews.

**Public Reviews:**

**Reviewer #1 (Public Review):**
Summary:The study used root tips from semi-hydroponic tea seedlings. The strategy followed sequential steps to draw partial conclusions.Initially, protoplasts obtained from root tips were processed for scRNA-seq using the 10x Genomics platform. The sequencing data underwent pre-filtering at both the cell and gene levels, leading to 10,435 cells. These cells were then classified into eight clusters using t-SNE algorithms. The present study scrutinised cell typification through protein sequence similarity analysis of homologs of cell type marker genes. The analysis was conducted to ensure accuracy using validated genes from previous scRNA-seq studies and the model plant *Arabidopsis thaliana*. The cluster cell annotation was confirmed using in situ RT-PCR analyses. This methodology provided a comprehensive insight into the cellular differentiation of the sample under study. The identified clusters, spanning 1 to 8, have been accurately classified as xylem, epidermal, stem cell niche, cortex/endodermal, root cap, cambium, phloem, and pericycle cells.Then, the authors performed a pseudo-time analysis to validate the cell cluster annotation by examining the differentiation pathways of the root cells. Lastly, they created a differentiation heatmap from the xylem and epidermal cells and identified the biological functions associated with the highly expressed genes.Upon thoroughly analysing the scRNA-seq data, the researchers delved into the cell heterogeneity of nitrate and ammonium uptake, transport, and nitrogen assimilation into amino acids. The scRNA-seq data was validated by *in situ* RT-PCR. It allows the localisation of glutamine and alanine biosynthetic enzymes along the cell clusters and confirms that both constituent the primary amino acid metabolism in the root. Such investigation was deemed necessary due to the paramount importance of these processes in theanine biosynthesis since this molecule is synthesised from glutamine and alanine-derived ethylamine.Afterwards, the authors analysed the cell-specific expression patterns of the theanine biosynthesis genes, combining the same molecular tools. They concluded that theanine biosynthesis is more enriched in cluster 8 "pericycle cells" than glutamine biosynthesis (Lines 271-272). However, the statement made in Line 250 states that the highest expression levels of genes responsible for glutamine biosynthesis were observed in Clusters 1, 3, 4, 6, and 8, leading to an unclear conclusion.

Thank you for your interest in and feedback on the paper. We have made revisions to the manuscript as per your suggestions. We would like to emphasize that the precursors of theanine biosynthesis are alanine-derived ethylamine and glutamate, not glutamine. Furthermore, in terms of the intermediates, only ethylamine is specific to the theanine biosynthetic pathway, as glutamate is the primary product of nitrogen assimilation and serves as a precursor for the biosynthesis of amino acids, proteins, chlorophyll, and many secondary metabolites.

In this study, we observed a high expression of genes encoding enzymes involved in the glutamate biosynthetic pathway (CsGOGATs and CsGDHs) across all 8 clusters, with particularly strong expression in cluster 1, 3, 4, 6, and 8 (Figure 4D and 5B). However, the gene encoding CsTSI responsible for catalyzing theanine biosynthesis from glutamate and ethylamine was determined to be more enriched in cluster 8 (Figure 5B and 5C). Therefore, we concluded that theanine biosynthesis was more enriched in cluster 8, whereas glutamate biosynthesis was more broadly active in clusters 1, 3, 4, 6 and 8.

The regulation of theanine biosynthesis by the MYB transcription factor family is well-established. In particular, CsMYB6, a transcription factor expressed specifically in roots, has been to promote theanine biosynthesis by binding to the promoter of the TSI gene responsible for theanine synthesis. However, their findings indicate that CsMYB6 expression is present in Cluster 3 (SCN), Cluster 6 (cambium cells), and Cluster 1 (xylem cells) but not in Cluster 8 (pericycle cells), which is known for its high expression of CsTSI. Similarly, their scRNA-seq data indicated that CsMYB40 and CsHHO3, which activate and repress CsAlaDC expression, respectively, did not show high expression in Cluster 1 (the cell cluster with high CsAlaDC expression). Based on these findings, the authors hypothesised that transcription factors and target genes are not necessarily always highly expressed in the same cells. Nonetheless, additional evidence is essential to substantiate this presumption.

Thank you for your advice. We fully agree that additional evidence is essential to support the presumption that transcription factors and target genes are not always highly expressed in the same cells. Therefore, in this study, we identified another transcription factor, CsLBD37, which was characterized to negatively regulate *CsAlaDC* expression in response to nitrogen levels. Consistent with our presumption, the expression of *CsLBD37* was not enriched in cluster 1, where the expression of *CsAlaDC* was primarily enriched (Figure 5B and 6D; Line 365).

To further identify supporting evidence, we also analyzed the expression of some transcription factors and their target genes in the model plant *Arabidopsis*, using published single cell RNA-seq data (Ryu et al., 2019; Wendrich et al., 2020; Zhang et al., 2019; Denyer et al., 2019; Jean-Baptiste et al. 2019; Shulse et al., 2019; Shahan et al., 2022) and database (Root Cell Atlas, https://rootcellatlas.org/; BAR, https://bar.utoronto.ca/#GeneExpressionAndProteinTools). Similar to the situation in tea plants, the regulators were not exactly the same as the cell types in which their target genes were highly expressed. For example, *AtARF7* and *AtARF19* were highly expressed in the cortex and stele, respectively, whereas their target genes *AtLBD16* and *AtLBD29* were highly expressed in endodermal cells (Okushima et al.,2007; Supplemental figure 8B and 8C; Line 312-325 and Line 525-526); *AtPHR1* was highly expressed in root epidermal cells and pericyte cells, but its target gene *AtF3’H* was highly expressed in the cortex and *AtRALF23* was highly expressed in xylem cells (Liu et al., 2022; Tang et al., 2022; Supplemental figure 8B and 8C; Line 322-327 and Line 527-530).

At the same time, we discussed that we cannot rule out the possibility of transcription factors regulating their target genes in the same cell type and both being highly expressed. One of the reasons is that these theanine-associated genes are promiscuous, having many target genes and regulate multiple biological processes in tea plants. We have only shown that high expression in the same cell type is not a necessary condition (Line 534-554). We strongly agree with the reviewer's opinion that more evidence is needed to illustrate this model in the future.

Reference:

Denyer, T. et al. (2019). Spatiotemporal developmental trajectories in the arabidopsis root revealed using high-throughput single-cell RNA sequencing. Dev Cell. **48**:840-852.e5.

Liu, Z. et al. (2022). PHR1 positively regulates phosphate starvation-induced anthocyanin accumulation through direct upregulation of genes *F3'H* and *LDOX* in *Arabidopsis*. Planta. **256**:42.

Okushima, Y. et al. (2007). ARF7 and ARF19 regulate lateral root formation via direct activation of *LBD/ASL* genes in *Arabidopsis*. Plant Cell. **19**:118-30.

Ryu, K. H., Huang, L., Kang, H. M. & Schiefelbein, J. (2019). Single-cell RNA sequencing resolves molecular relationships among individual plant cells. Plant Physiol. **179**:1444-1456.

Shahan, R. et al. (2022). A single-cell Arabidopsis root atlas reveals developmental trajectories in wild-type and cell identity mutants. Dev Cell. **57**:543-560.e9.

Shulse, C. et al. (2019). High-throughput single-cell transcriptome profiling of plant cell types. Cell Rep. **27**:2241-2247.e4.

Tang, J. et al. (2022). Plant immunity suppression via PHR1-RALF-FERONIA shapes the root microbiome to alleviate phosphate starvation. EMBO J. **41**:e109102.

Wendrich, J.R., et al. (2020). Vascular transcription factors guide plant epidermal responses to limiting phosphate conditions. Science. **370**:eaay4970.

Zhang, T. et al. (2019). A single-cell RNA sequencing profiles the developmental landscape of arabidopsis root. Mol Plant. **12**:648-660.

Lastly, the authors have discovered a novel transcription factor belonging to the Lateral Organ Boundaries Domain (LBD) family known as CsLBD37 that can co-regulate the synthesis of theanine and the development of lateral roots. The authors observed that CsLBD37 is located within the nucleus and can repress the CsAlaDC promoter's activity. To investigate this mechanism further, the authors conducted experiments to determine whether CsLBD37 can inhibit CsAlaDC expression in vivo. They achieved this by creating transiently CsLBD37-silenced or over-expression tea seedlings through antisense oligonucleotide interference and generation of transgenic hairy roots. Based on their findings, the authors hypothesise that CsLBD37 regulates CsAlaDC expression to modulate the synthesis of ethylamine and theanine.Additionally, the available literature suggests that the transcription factors belonging to the Lateral Organ Boundaries Domain (LBD) family play a crucial role in regulating the development of lateral roots and secondary root growth. Considering this, they confirmed that pericycle cells exhibit a higher expression of CsLBD37. A recent experiment revealed that overexpression of CsLBD37 in transgenic *Arabidopsis thaliana* plants led to fewer lateral roots than the wild type. From this observation, the researchers concluded that CsLBD37 regulates lateral root development in tea plants. I respectfully submit that the current conclusion may require additional research before it can be considered definitive.Further efforts should be made to investigate the signalling mechanisms that govern CsLBD37 expression to arrive at a more comprehensive understanding of this process. In the context of Arabidopsis lateral root founder cells, the establishment of asymmetry is regulated by LBD16/ASL18 and other related LBD/ASL proteins, as well as the AUXIN RESPONSE FACTORs (ARF7 and ARF19). This is achieved by activating plant-specific transcriptional regulators such as LBD16/ASL18 (Go et al., 2012, https://doi.org/10.1242/dev.071928). On the other hand, other downstream homologues of LBD genes regulated by cytokinin signalling play a role in secondary root growth (Ye et al., 2021, https://doi.org/10.1016/j.cub.2021.05.036). It is imperative to shed light on the hormonal regulation of CsLBD37 expression in order to gain a comprehensive understanding of its involvement in the morphogenic process.

We are very grateful for your valuable suggestions and we fully agree with you. In an earlier study, we also observed a link between theanine metabolism, hormone metabolism and root development (Chen et al., 2022), but there is still insufficient evidence to fully characterize these links. In the current study, the focus was on the cell-specific theanine biosynthesis, transport and regulation, and we identified that CsLBD37 negatively regulates theanine biosynthesis. However, the upstream regulatory mechanism of CsLBD37 has not been addressed in this study. It is a pertinent question for future investigation as to how CsLBD37 is regulated in root development. We have included the following additional discussion in the revised manuscript: “Besides, it has been reported that LBD family TFs were regulated by, or interacted with, regulators of hormone pathways (e.g., ARFs) to regulate the process of root morphogenesis (Goh et al., 2012; Ye et al., 2021). Based on these findings, we speculated that CsLBD37 is likely regulated by, or interacts with, other proteins to form a complex to regulate root development or theanine biosynthesis.” (Line 573-576). At the same time, we revised the text “These results provided support for a model in which CsLBD37 plays a role in regulating lateral root development in tea plants” to “These findings suggested that CsLBD37 may play a role in regulating lateral root development in tea plant roots” (Line 401-402).

Reference:

Chen, T. et al. (2022). Theanine, a tea plant specific non-proteinogenic amino acid, is involved in the regulation of lateral root development in response to nitrogen status. Hortic. Res. **10**:uhac267.

Goh, T., Joi, S., Mimura, T. & Fukaki, H. (2012). The establishment of asymmetry in *Arabidopsis* lateral root founder cells is regulated by LBD16/ASL18 and related LBD/ASL proteins. Development **139**:883-893.

**Y**e, L. et al. (2021). Cytokinins initiate secondary growth in the *Arabidopsis* root through a set of LBD genes. Curr. Biol. **31**:3365-3373.e3367.

Strength:The manuscript showcases significant dedication and hard work, resulting in valuable insights that serve as a fundamental basis for generating knowledge. The authors skillfully integrated various tools available for this type of study and meticulously presented and illustrated every step involved in the survey. The overall quality of the work is exceptional, and it would be a valuable addition to any academic or professional setting.Weaknesses:In its current form, the article presents certain weaknesses that need to be addressed to improve its overall quality. Specifically, the authors' conclusions appear to have been drawn in haste without sufficient experimental data and a comprehensive discussion of the entire plant. It is strongly advised that the authors devote additional effort to resolving the abovementioned issues to bolster the article's credibility and dependability. This will ensure that the article is of the highest quality, providing readers with reliable and trustworthy information.

Thank you for your feedback. We acknowledge that our experiments and data require further improvement. Currently, the genetic transformation of the tea plant remains a challenge, making it difficult to obtain sufficient *in vivo* evidence. Despite this situation, we have made every effort to obtain support for our conclusions based on the current situation and available technology. Indeed, additional studies will be performed once the impediment associated with genetic transformation of the tea plant has been resolved.

**Reviewer #2 (Public Review):**
Summary:In their manuscript, Lin et al. present a comprehensive single-cell analysis of tea plant roots. They measured the transcriptomes of 10,435 cells from tea plant root tips, leading to the identification and annotation of 8 distinct cell clusters using marker genes. Through this dataset, they delved into the cell-type-specific expression profiles of genes crucial for the biosynthesis, transport, and storage of theanine, revealing potential multicellular compartmentalization in theanine biosynthesis pathways. Furthermore, their findings highlight CsLBD37 as a novel transcription factor with dual regulatory roles in both theanine biosynthesis and lateral root development.Strengths:This manuscript provides the first single-cell dataset analysis of roots of the tea plants. It also enables detailed analysis of the specific expression patterns of the gene involved in theanine biosynthesis. Some of these gene expression patterns in roots were further validated through in-situ RT-PCR. Additionally, a novel TF gene CsLBD37's role in regulating theanine biosynthesis was identified through their analysis.Weaknesses:Several issues need to be addressed:(1) The annotation of single-cell clusters (1-8) in Figure 2 could benefit from further improvement. Currently, the authors utilize several key genes, such as CsAAP1, CsLHW, CsWAT1, CsIRX9, CsWOX5, CsGL3, and CsSCR, to annotate cell types. However, it is notable that some of these genes are expressed in only a limited number of cells within their respective clusters, such as CsAAP1, CsLHW, CsGL3, CsIRX9, and CsWOX5. It would be advisable to utilize other marker genes expressed in a higher percentage of cells or employ a combination of multiple marker genes for more accurate annotation.

Thank you for your comments. In this study, we first utilized classical marker genes, such as *CsWAT1* and *CsPP2*, to annotate cell types. The expression patterns of these marker genes were confirmed using *in situ* RT-PCR. Additionally, a combination of multiple marker genes was employed for cell type annotation. We also analyzed the top 10 cluster-enriched genes, in each cluster, and their homologous expression in Arabidopsis, populus, etc., to serve as a reference for cluster annotation (Figure 2D; Supplemental Figures 2-6; Supplemental data 3). Subsequently, differentiation trajectories of root cells were analyzed based on pseudo-time analyses, which aligned well with cell type annotation and further supported the reliability of our annotations through these combined methods.

(2) Figure 3 could enhance clarity by displaying the trajectory of cell differentiation atop the UMAP, similar to the examples demonstrated by Monocle 3.

Thanks for this advice. We have supplied the trajectory of cell differentiation atop the UMAP in the revised supplemental figure 7 (Line 185).

(3) The identification of CsLBD37 primarily relies on bulk RNA-seq data. The manuscript could benefit from elaborating on the role of the single-cell dataset in this context.

Thanks for your comments. In this study, we determined that *CsTSI* was highly expressed in cluster 8, but its regulator *CsMYB6* was highly expressed in cluster 3, cluster 6 and cluster 1 (Line 301-304). Thus, target genes and their regulators seem not to always be highly expressed in the same cell cluster. A similar situation was also observed in terms of *CsAlaDC* transcriptional regulation (Line 305-311). Based on these findings, we hypothesized that, for the regulation of theanine biosynthesis, it is not necessary for transcription factors and target genes to always be highly expressed in the same cells. Thus, taking the transcriptional regulation of *CsAlaDC* as an example, we next analyzed the TFs that were co-expressed with *CsAlaDC* to test this notion. We used scRNA-seq data to screen for genes that were not highly co-expressed with *CsAlaDC*, such as CsLBD37, to test our hypothesis (Line 338-340 and Line 365).

(4) The manuscript's conclusions predominantly rely on the expression patterns of key genes. This reliance might stem from the inherent challenges of tea research, which often faces limitations in exploring molecular mechanisms due to the lack of suitable genetic and molecular methods. The authors may consider discussing this point further in the discussion section.

Thanks for your suggestions and we totally agree. We discussed this point in the discussion section, “In some non-model plants, including tea, transgenic technologies are not currently available and, hence, we used *in situ* RNA hybridization to establish the location(s) for gene expression. In some studies, isolation of different cell types was combined with q-RT-PCR to detect cell-type marker gene expression (Wang et al., 2022). However, this approach has two limitations in that it cannot display the gene location directly and has only low resolution”, “After numerous trials, we were able to optimize *in situ* RT-PCR assays (detailed in the Methods), which enabled a cell-specific characterization of gene expression in tea plant root cells, prior to establishing a genetic transformation system for tea…we note the challenge associated with weak calling of homologous marker genes…” (Line 431-444).

**Reviewer #3 (Public Review):**
Summary:Lin et al., performed a scRNA-seq-based study of tea roots, as an example, to elucidate the biosynthesis and regulatory processes for theanine, a root-specific secondary metabolite, and established the first map of tea roots comprised of 8 cell clusters. Their findings contribute to deepening our understanding of the regulation of the synthesis of important flavor substances in tea plant roots. They have presented some innovative ideas.It is notable that the authors - based on single-cell analysis results - proposed that TFs and target genes are not necessarily always highly expressed in the same cells. Many of the important TFs they previously identified, along with their target genes (CsTSI or CsAlaDC), were not found in the same cell cluster. Therefore, they proposed a model in which the theanine biosynthesis pathway occurs via multicellular compartmentation and does not require high co-expression levels of transcription factors and their target genes within the same cell cluster. Since it is not known whether the theanine content is absolutely high in the cell cluster 1 containing a high CsAlaDC expression level (due to the lack of cell cluster theanine content determination, which may be a current technical challenge), it is difficult to determine whether this non-coexpressing cell cluster 1 is a precise regulatory mechanism for inhibiting theanine content in plants.

Thank you for your comments. We concur with your assessment that the accumulation level of the spatial distribution of theanine may affect the expression of these genes. However, as you said, due to some technical limitations, we are not currently in a position to verify this distribution of theanine at the root cell spatial level. The spatial distribution of theanine in the roots can be affected by transport processes. So, it is likely that the cell types in which theanine is distributed do not exactly correspond to the cell types in which theanine is being synthesized (Line 491-493). We will make efforts in this direction to characterize the spatial distribution of theanine using techniques such as spatial metabolome and mass spectrometry imaging in the future (Line 582-586).

In fact, there are a small number of cells where TFs and *CsAlaDC* are simultaneously highly expressed, but the quantity is insufficient to form a separate cluster. However, these few cells may be sufficient to meet the current demands for theanine synthesis. This possibility may better align with some previous experiments and validation results in this study. Moreover, I feel that under normal conditions, plants may not mobilize a large number of cells to synthesize a particular substance. Perhaps, cell cluster 1 is actually a type of cell that inhibits the synthesis of theanine, aiming to prevent excessive theanine production? I do not oppose the model proposed by the author, but I feel there is a possibility as I mentioned. If it seems reasonable, the author may consider adding it to an appropriate position in the discussion.

Thanks a lot for your suggestion. We agree that tea plant roots likely have mechanisms aiming to prevent excessive theanine production.We have improved our discussion according your suggestion.

Theanine is the most abundant free amino acid in the tea plant, accounting for 1-2% of leaf dry weight (Line 62-63), and can even reach 4-6% in the root, accounting for more than 60%-80% of the total free amino acids (Yang et al., 2020). This means that theanine biosynthesis indeed requires the root cells to consume significant resources and energy. Thus, theanine biosynthesis needs to be controlled by a series of regulation mechanisms, which would function as a “brake”. In a previous study, we suggested that CsMYB40 and CsHHO3 bound to the *CsAlaDC* promoter to regulate theanine synthesis, at the transcription level, in “accelerator” or “brake” mode to maintain stable synthesis of theanines (Guo et al., 2022). At a posttranslational level, CsTSI and CsAlaDC are modified by ubiquitination, which is probably involved in the degradation of these proteins in response to N levels (Wang et al., 2021). In the current study, we discovered a novel “brake” in the form of spatial separation. The differential expression of *AlaDC* and *TSI* suggests that ethylamine and theanine are synthesized in separate different cell types, allowing cell compartmentalization of the synthetic precursor and the product to form multicellular compartmentation of metabolites (Line 270-280). On the one hand, compartmentalization may effectively prevent interference between secondary metabolic pathways, whereas compartmentalization could also be used as a way of metabolic regulation to avoid excessive, or inhibition of, theanine synthesis (Line 483-488).

Reference

Guo, J. et al. (2022). Potential “accelerator” and “brake” regulation of theanine biosynthesis in tea plant (*Camellia sinensis*). Hortic. Res. **9**:uhac169.

Yang, T. et al. (2020). Transcriptional regulation of amino acid metabolism in response to nitrogen deficiency and nitrogen forms in tea plant root (*Camellia sinensis* L.). Sci. Rep. **10**:6868.

Wang, Y. et al. (2021). Nitrogen-Regulated Theanine and Flavonoid Biosynthesis in Tea Plant Roots: Protein-Level Regulation Revealed by Multiomics Analyses. J Agric Food Chem. **69**:10002-10016.

**Recommendations for the authors:**

**Reviewer #2 (Recommendations For The Authors):**
(1) The dataset, including the raw sequencing data and processed files is *.Rdata and should be deposited in a public database for accessibility and reproducibility.

Thanks for your comments and advice. The raw data and processed files have been submitted to the Gene Expression Omnibus (https://www.ncbi.nlm.nih.gov/geo/) under accession number GSE267845 (Line 763-764).

(2) Providing the code for the primary analysis steps in a publicly accessible location would facilitate others in replicating the analysis.

Thank you for your comment. Unfortunately, we have been unable to obtain permission to publicly release a portion of the primary analysis code due to its intellectual property belonging to OE Corporation.

(3) Enhancements in the writing of the manuscript are recommended for improved clarity and coherence.

Thanks. We revised our writing to improve the manuscript clarity and coherence.

**Reviewer #3 (Recommendations For The Authors):**
Suggestions for revisions:(1) Introduction and Discussion, there are too many paragraphs, even one sentence is a paragraph. I suggest that all the sentences in Introduction be merged into three big paragraphs. For example, lines 30-57 become the first paragraph, lines 58-87 become the second paragraph, lines 88-106 become the third paragraph, and the authors can merge them reasonably according to the content. The discussion part is also suggested to be divided into several paragraphs according to the focus, and perhaps it is more appropriate to give a title to each paragraph.

Thank you for your comments and suggestions. We have merged several paragraphs and added a title to each paragraph in the Discussion section (“Cell cluster annotation of non-transgenic plants” in line 428; “Nitrogen metabolism and transport of tea plant root at the single cell level” in line 445; “Multicellular compartmentation of theanine metabolism and transport” in line 469; “The regulation of theanine biosynthesis at the single cell level” in line 517; “Cross-talk between theanine metabolism and root development” in line 554).

(2) Tea is a food, while tea tree is a substance. It should be tea plant root instead of tea root, it is suggested to revise this issue in the whole text.

Thanks. We corrected “tea root” to “tea plant root” in this manuscript.

(3) Lines 35-43, this sentence is too long, suggest each example should be one sentence.

Thanks. We revised this sentence into short sentences. We changed this part to “Root-synthesized flavonoids regulate root tip growth through affecting auxin transport and metabolism (Santelia et al., 2008; Wan et al., 2018). Legume roots secrete flavonoids as signaling agents to attract symbiotic bacteria, such as *Rhizobium* for nitrogen fixation (Hartman et al., 2017). In *Abies nordmanniana*, volatile organic compounds (e.g., propanal, g-nonalactone, and dimethyl disulfide) function to recruit certain bacteria or fungi, such as *Paenibacillus*. *Paenibacillus* sp. S37 produces high quantities of indole-3-acetic acid that can then promote plant root growth (Garcia-Lemos et al., 2020; Schulz-Bohm et al., 2018).” (Line 35-42)

(4) Line 510 is missing a reference.

Thank you - we have added the reference in the revised manuscript (Line 549 and Line 840-842).